

# Development and validation of a supervised machine learning radar Doppler spectra peak finding algorithm

Heike Kalesse[1,2], Teresa Vogl[1,2], Cosmin Paduraru[3], and Edward Luke[4]

[1]Leibniz Institute for Tropospheric Research, Leipzig, Germany
[2]Institute for Meteorology, Universität Leipzig, Leipzig, Germany
[3]Department of Mining and Materials Engineering, McGill University, Montreal, Canada
[4]Environmental and Climate Sciences Department, Brookhaven National Laboratory, Upton, New York

**Correspondence:** Heike Kalesse (heike.kalesse@uni-leipzig.de)



**Abstract.** In many types of clouds, multiple hydrometeor populations can be present at the same time and height. Studying the evolution of these different hydrometeors in a time-height perspective can give valuable information on cloud particle composition and microphysical growth processes. However, as a prerequisite, the number of different hydrometeor types in a certain cloud volume needs to be quantified. This can be accomplished using cloud radar Doppler velocity spectra from profiling

cloud radars if the different hydrometeor types have sufficiently different terminal fall velocities to produce individual Doppler spectrum peaks. Here we present a newly developed supervised machine learning radar Doppler spectra peak finding algorithm (named PEAKO). In this approach, three adjustable parameters (spectrum smoothing span, prominence threshold, and minimum peak width at half-height) are varied to obtain the set of parameters which yields the best agreement of user-classified and machine-marked peaks. The algorithm was developed for Ka-band ARM Zenith-pointing Radar (KAZR) observations

obtained in thick snow fall systems during the Atmospheric Radiation Measurement Program's (ARM) mobile facility AMF2 deployment at Hyytiälä, Finland, during the Biogenic Aerosols – Effects on Clouds and Climate (BAECC) field campaign. The performance of PEAKO is evaluated by comparing its results to existing Doppler peak finding algorithms. The new algorithm is found to perform well. Its advantage is that the detected features are less noisy and more consistent in time and height than the peak finding results of other algorithms. In the future, the PEAKO algorithm will be adapted to other cloud radars and other

types of clouds consisting of multiple hydrometeors in the same cloud volume.

## 1 Introduction

Determining cloud composition in terms of hydrometeor populations is a non-trivial task in thick cold precipitating clouds below 0°C. In these clouds, supercooled liquid water droplets and solid ice crystals of a variety of shapes and sizes can coexist at temperatures between -40°C and 0°C. Mixed-phase clouds and thick cold precipitating cloud systems play an important role

in the Earth's climate, due to their strong influence on the radiative budget (Tan et al., 2016). Global climate models (GCM) still have problems in representing mixed-phase clouds, and especially the supercooled liquid fraction (SLF) accurately (Komurcu et al., 2014).

This motivates the need for highly time- and range-resolved observations of the occurrence of different hydrometeor populations and of cloud phase in the vertical column. The first step towards this characterization of hydrometeor types is the

determination of the number of different populations within a certain cloud volume. Profiling cloud Doppler radars are well suited for this task for two reasons:

(i) They are able to penetrate the complete atmospheric column (except for strongly precipitating deep convective clouds), i.e. also beyond the range where lidar is fully attenuated, and

(ii) they can be used as a stand-alone means of inferring the number of different hydrometeor populations and in certain cir-

cumstances even cloud phase, because different ice particle populations (and sometimes liquid cloud droplets) and ice particles, which are present simultaneously within a radar sampling volume, are characterized by different terminal fall velocities due to their different particle size distributions and densities.

Each of these different particle size distributions thus generates a peak in the radar Doppler velocity spectrum (Kollias et al.,



2016). However, sub-volume turbulence broadens the cloud Doppler spectra peaks and thus smears/smoothes the microphysical signature. Using narrow-beam width antennas and optimizing observational strategies with short dwell time and high vertical resolution reduces turbulence-induced spectrum broadening (Kollias et al., 2016). However, the observed Doppler spectrum is always a convolution of microphysical and dynamical effects.

In order to infer microphysical properties from the radar Doppler spectrum, the peaks have to be separated. Since spectra can be noisy and peaks can be merged, this is a non-trivial task, which has already been approached in multiple ways in the past for different cloud types: Shupe et al. (2004) were able to separate observed Doppler velocity spectra into a liquid and an ice spectral mode for a 30-minute long altostratus case study. They empirically defined criteria, which were applied by an algorithm to distinguish multiple peaks in the radar Doppler spectra.

The Microscale Active Remote Sensing of Clouds (MicroARSCL) data product (Kollias et al., 2007; Luke et al., 2008) is generated by a post-processing routine applied to Doppler spectra recorded by the U.S. Department of Energy (DOE)'s Atmospheric Radiation Measurement (ARM) Program millimeter wavelength cloud radars. It uses the morphology of the Doppler spectrum to determine shape parameters like skewness and kurtosis for both the primary peak (highest reflectivity) and, if applicable, an additional noise-separated secondary peak (of lower reflectivity). The peak power densities and modal

velocities of up to two local maxima (sub-peaks) located within the primary peak are also included. The MicroARSCL product has e.g. been used by Riihimaki et al. (2016) and Oue et al. (2018). The former used it to infer hydrometeor phase in a tropical deep convective system, the latter to study hydrometeor populations in deep precipitating systems in the Arctic. Oue et al. (2018) found multimodal Doppler spectra in the dendritic/planar growth layer as well as in mixed-phase layers. They also highlighted the added value of joint analysis of Doppler spectra and polarimetric variables from scanning cloud radar

observations for snow microphysical studies.

Verlinde et al. (2013) were able to separate four hydrometeor populations (background ice, cloud, drizzle and new ice) using continuity of spectral modes in time and height in an Arctic cloud case study, in combination with high spectral resolution lidar (HSRL) and in-situ observations.

Kalesse et al. (2016) presented a case study of the BAECC field campaign, in which frontal snow was falling through a

supercooled liquid water layer (SWL), which lead to riming of the snow flakes. They could distinguish up to three noise-floor separated peaks in the recorded radar Doppler spectra, which were then used to track microphysical processes along slanted fall streaks. However, it is stressed that this case was special due to the separation of the peaks by the noise-floor. Usually, merged peaks are observed, motivating the need for the development of robust cloud radar Doppler spectrum peak separation techniques.

Williams et al. (2018) identify multiple Doppler peaks using the depth of the local minimum between main peak and sub-peak as separation criteria for KAZR observations of liquid-only and mixed-phase clouds at Oliktok Point, Alaska.

All these efforts, using in part considerably differing approaches, show that there is a need to correctly separate multiple merged peaks in Doppler spectra to aid microphysical understanding of mixed phase cloud processes as well as to improve hydrometeor classification techniques. At the same time, recent studies highlight the role of machine learning as a tool for

hydrometeor classification based on remote sensing data e.g., Besic et al. (2016); Praz et al. (2017).



## 2   Data set description

The Biogenic Aerosols-Effects on Clouds and Climate (BAECC; Petäjä et al., 2016) campaign took place at the Station for Measuring Ecosystem-Atmosphere Relations II (SMEARII) in Hyytiälä, Finland (61°51′N, 24°17′E, 150 m above sea level). The ARM Program deployed their Second ARM Mobile Facility (AMF2) from February to September 2014. Within this frame,

a snowfall experiment (BAECC SNEX) took place as a collaborative effort between DOE ARM, University of Helsinki, the Finnish Meteorological Institute (FMI), the National Aeronautics and Space Administration (NASA) and Colorado State University. An intensive operation period (IOP) from 1 February to 30 April 2014 was aimed at measuring snowfall microphysics using a comprehensive suite of remote sensing instruments, complemented by surface-based precipitation observations.

The AMF is constituted of several ground-based remote sensing instruments, including among other things a 35 GHz Ka-band

ARM Zenith-pointing Radar (KAZR), as well as a W-, Ka-, and X-band Scanning ARM Cloud Radar (Kollias et al., 2014), a High Spectral Resolution Lidar (HSRL) and a micropulse lidar (MPL). Supplementing these measurements, radiosondes were launched four times daily. This study will focus on the Doppler spectra recorded by the KAZR as main data basis, and will utilize other observations (ground-based in-situ, HSRL - if applicable) for comparison and validation purposes. The KAZR was operated with a temporal resolution of 2 s, a vertical range gate spacing of 30 m, and a Doppler velocity spectrum resolution

(bin width) of 2.37 cm s$^{-1}$.

## 3   Methodology

### 3.1   PEAKO algorithm description

In this study, a supervised Doppler spectra peak finding algorithm (in the following text referred to as PEAKO) was developed, which is trained via hand-marked Doppler peaks as input. The learning process is split into two phases, the training phase and

the test phase, respectively. For that purpose, three data sets, each containing example input and the corresponding desired output, are created:

   – a first training data set, used to obtain an initial model

   – a second training data set, about half as large as the first training data set, used to tune the model

   – a testing data set, which has approximately the same size as the second training data set and is used for model evaluation

With the help of a graphical Matlab interface, in which the currently to-be-marked Doppler spectrum and its surrounding neighbors (in time and height) are displayed in logarithmic space (see Fig. 1), pronounced Doppler spectrum peaks are hand-marked by an experienced user. Even though this approach is subjective, criteria such as peak width, dynamic range, i.e. the height above noise floor, skewness of the spectrum and consistency of the feature (peak) in time-height are taken into account. The locations of these hand-marked peaks (in mean Doppler velocity ($V_D$) units [m s$^{-1}$]), as well as their corresponding signal

powers (in dBZ) are then saved as data matrices.





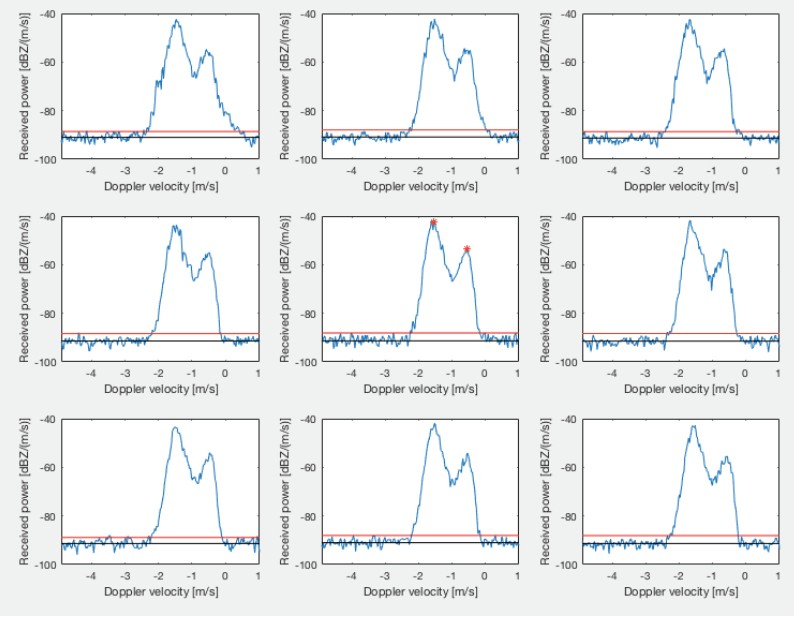

**Figure 1.** Example of graphical user interface for peak-marking by hand. For the Doppler spectrum in the center, two peaks (red stars) were marked by the user (HK). The surrounding spectra display the neighboring spectra. Data: random spectra of KAZR observed at TMP on February 16, 2014, 0.03-0.05 UTC between 1-1.2 km height. The red line marks the maximum noise floor, the black line the mean spectral noise, determined according to Hildebrand and Sekhon (1974).

The training and test data sets were chosen as individual non-overlapping time-height areas rather than randomly splitting all hand-marked spectra into training and test categories. For the training phase, data recorded on 21 February 21 2014, 22-23 UTC were utilized, a time period, which was studied in greater detail before (Kalesse et al., 2016; Mason et al., 2018). Data from 16 February 2014, 0-1 UTC, which were in part investigated in a case study presented in Kneifel et al. (2015), are

used in the training phase as well. The third case selected for the training data set, 21 February 2014, 23-24 UTC, was in part analyzed by Kneifel et al. (2015) as well. The test set is comprised of two 1-hour-cases, which were recorded on 2 February 2014, 16-17 UTC, and on 7 February 2014, 23-24 UTC, respectively. In the case of 2 February 2014, 16-17 UTC, the study area was set within the lowest liquid layer where independent lidar (HSRL) measurements can be used to check the performance of the PEAKO algorithm for liquid-peak detection. Unfortunately, the HSRL was fully attenuated by near-surface liquid-layers

during the other case studies. The chosen period on 7 February 2014 overlaps with another case investigated by Kneifel et al. (2015). Table 1 gives a summary which measurement periods were used for which of the data sets.

The PEAKO algorithm includes a set of three subsequently described adjustable parameters (smoothing span, prominence threshold, minimum peak width at half-height), which are varied to obtain the set of parameters which yields the best agreement of hand-marked and machine-marked peaks. The search for the best parameter combination is done via search through a

finite set of values for the three-dimensional search space.



**Table 1.** Overview of the measurement periods used in training and test data sets containing hand-marked peaks. Published studies of the selected periods, to which results can be compared, are noted as well.

|  | training set | test set |
|---|---|---|
| 2014-02-02 |  | X |
| 16 - 17 UTC |  | (comparison to HSRL) |
| 2014-02-07 |  | X |
| 23-24 UTC |  | Kneifel et al. (2015) |
| 2014-02-16 | X |  |
| 0-1 UTC | Kneifel et al. (2015) |  |
| 2014-02-21 | X |  |
| 22-23 UTC | Kalesse et al. (2016) |  |
| 2014-02-21 | X |  |
| 23-24 UTC | Kneifel et al. (2015) |  |

The span for smoothing defines the fraction of the total number of data points (here: Doppler bins) of one Doppler velocity spectrum to be considered for spectral smoothing. Spectral smoothing is performed using local regression using weighted linear least squares and a 2nd degree polynomial model (loess). This smoothing method was chosen empirically after testing different methods since it showed the most promising results. The span is varied in a range between 3.5% and 13%, regularly

spaced with a distance of 0.5%. Spectral smoothing is performed on an average of 16 s temporal and 90 m spatial dimension, i.e. for the given KAZR time-height resolution of 2 s (time) and 30 m (range), the mean of 4 neighbors in time and 2 neighbors in height-dimension was made before the machine-based peak finding was applied to smooth out some spurious features in individual Doppler spectra. With hydrometeor populations usually appearing in distinct layers which are persistent over a certain period of time, more neighbors in time than height are used for averaging.

The prominence threshold is a measure of how much a peak stands out relative to the other peaks in the considered Doppler spectrum. The prominence of a peak is the power difference (dynamic range) of the peak's maximum and the signal's minimum between the considered peak and the nearest higher peak. Concerning the highest peak of the Doppler spectrum, the prominence is the power difference between the peak maximum and the noise floor. This parameter is varied between 0 and $2\,\mathrm{dBZ\,m^{-1}\,s^{-1}}$ in the training phase of PEAKO. Fig. 2 illustrates the definition of the peak prominence: A spectrum with three merged peaks is

shown and their prominences are drawn as red vertical lines. In the case of the rightmost peak ($\mathrm{V}_D \approx -0.5\,\mathrm{m\,s^{-1}}$), the prominence is defined as the power difference between the peak's maximum and the minimum between this peak and the nearest higher peak. This minimum is located approximately at the leftmost peak's right edge (marked with a solid black vertical line at around $-1.1\,\mathrm{m\,s^{-1}}$). For the peak with the lowest power at $\mathrm{V}_D \approx -0.8\,\mathrm{m\,s^{-1}}$, the prominence of $0.25\,\mathrm{dBZ\,m\,s^{-1}}$ is barely visible because it is defined as only the distance between this peak's maximum and the minimum to the closest higher peak,

which is the peak with the lowest absolute $\mathrm{V}_D$.

The third adjustable parameter is the minimum peak width at half-height. The range of values in which it is varied (4.2 to 8.4



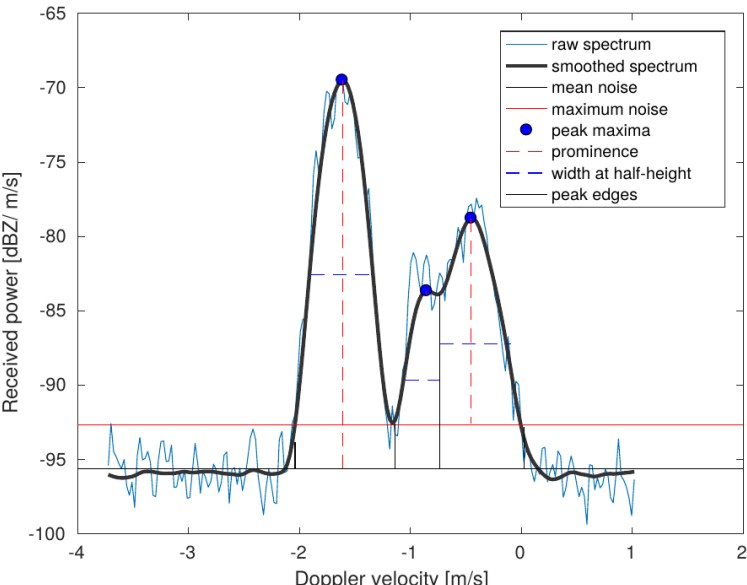

**Figure 2.** Example spectrum with multiple merged peaks, recorded on 21 February 21 2014, 22.7 UTC in 2.44 km altitude. The smoothed spectrum (average of neighbor spectra in time and height domain, smoothed using a span of 8.5%) is shown as well (bold black line). For each of the three peaks marked in this spectrum (blue dots), the prominences (red dashed lines) and widths at half height (blue dashed lines), as well as the edges (vertical black lines) of the peaks are marked.

Doppler velocity bins, spaced with a distance of 1.05) was determined from a low-turbulence cloud region only consisting of liquid droplets (21 February 2014, 22.53 - 22.59 UTC 2.9-3.1 km). Doppler spectra peaks in low-turbulent liquid cloud droplet layers are very narrow and thus suited to determine the minimum width of a peak considered as physically meaningful. At the given KAZR resolution, these peaks were found to be between 4.2 and 8.4 $V_D$ bins wide corresponding to about 10 -
20 cm s$^{-1}$.

 To determine the optimal parameter combination, a similarity measure is defined, based on the maximum overlapping area of detected peaks as illustrated in Fig 3: For a certain set of smoothing span, minimum peak width and prominence threshold, the algorithm will detect certain peaks in a Doppler spectrum (shown as red circles in Fig. 3). For these peaks, as well as for the peaks marked by a user in the same Doppler spectrum (red stars in Fig. 3), the edges (marked with vertical lines) are
determined, which are defined as follows: The edge is either the first Doppler bin, where the spectrum power is smaller than the maximum noise floor, or, in case of a merged peak, the minimum (saddle point) between the merged peaks. In the next step, the overlapping area for each pair of hand-marked and machine-found peaks is identified: In case of multiple peaks in one spectrum, the sum of all overlapping areas is determined. Non-overlapping regions caused either by a mismatch in the number of hand-marked and machine-found peaks or by a different location (in x-direction, i.e., in Doppler velocity) of the pair of
peaks, are penalized in the similarity area measure by subtracting the non-overlapping area (marked red hatched in Fig. 3) of





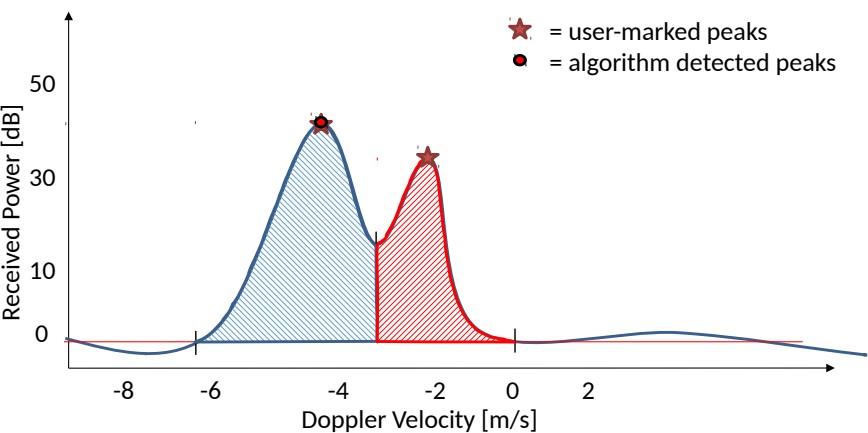

**Figure 3.** Schematic to visualize how the similarity measure to compare user-marked and algorithm-found peaks in Doppler spectra: Areas of matching peaks are summed up (blue hatched area), and the areas of mismatched peaks (red hatched) are subtracted.

the mismatched peaks. The optimum parameter combination is the triplet of span, prominence threshold and minimum peak width, for which the similarity has its maximum value.

### 3.2 Description of other radar Doppler spectra peak-finding algorithms

Three other radar Doppler velocity spectrum peak finding algorithms were compared to PEAKO and will be briefly explained in the following.

The algorithm described in Shupe et al. (2004) (from now on referred to as "Shupe_04") uses peak finding criteria optimized to one relatively short (30 min) mixed-phase cloud case study period. In short, the power of the primary (strongest) peak must be at least 4 standard deviations of the noise above the mean spectral noise level. In addition, one or more secondary peaks are

10 identified as maxima being at least 2.5 standard deviations of the noise greater than the mean spectral noise level. Both primary and secondary peaks must have a width of at least $0.448 \, \mathrm{m\,s^{-1}}$. Merged peaks are identified as separate spectral modes if the saddle point between the two maxima is lower than 65% of the lowest of the two peaks from the noise level.

The MicroARSCL data product (Kollias et al., 2007; Luke et al., 2008) decomposes noise floor subtracted and 3-bin-boxcar-filtered radar Doppler spectra into a primary peak (defined as the peak containing the bin with maximum spectral power

density), dominating the total reflectivity and containing up to two sub-peaks, and a possible secondary peak. Fixed thresholds, i.e. minimum primary or secondary peak width ($\mathrm{Pw}_{min}$), minimum sub-peak height ($\mathrm{Ph}_{min}$), minimum sub-peak separation ($\mathrm{Ps}_{min}$), and minimum primary-secondary peak edge separation ($\mathrm{Pn}_{min}$), are applied to extract a suite of variables from the Doppler spectra. $\mathrm{Pw}_{min}$ is set to 5 Doppler velocity bins ($0.12 \, \mathrm{m\,s^{-1}}$), $\mathrm{Ps}_{min}$ to 3 Doppler velocity bins ($0.07 \, \mathrm{m\,s^{-1}}$), $\mathrm{Pn}_{min}$





is 1 Doppler velocity bin ($0.0237\,\mathrm{m\,s^{-1}}$), and $\mathrm{Ph}_{min}$ is $1\,\mathrm{dBZ\,m^{-1}\,s^{-1}}$. For the comparative study, a third technique, being a polynomial fitting algorithm as described in Kollias et al. (1999) and Kollias et al. (2003) was also applied to the radar Doppler spectra which are analyzed in this study. This routine first extracts the parts of the spectrum above the maximum noise floor (signal-to-noise threshold determined by Hildebrand and Sekhon (1974)) and extends the edges of the found peaks down to

the mean noise floor. In a next step, each continuous sample of data above the noise floor is identified as a sub-spectrum. Sub-spectra that are classified as being too narrow (with velocity ranges of the peak smaller than $0.2\,\mathrm{m\,s^{-1}}$) are excluded. For each of the remaining sub-spectra, polynomial fitting of 12th order is applied. The first and second derivatives are taken to identify minima and maxima. Peaks are defined as sequences of minimum-maximum-minimum. Peaks having a velocity range smaller than $0.2\,\mathrm{m\,s^{-1}}$ are ignored, as well as peaks with an amplitude smaller than $2\,\mathrm{dB}$, with amplitude being defined as the

difference in reflectivity between consecutive minimum and maximum.

## 4    Results and Discussion

The following chapter will be structured as follows: In section 4.1, the best parameter values obtained during the training phase of the PEAKO algorithm are summarized. The peaks detected by one of these best parameter combinations are compared to peaks found by Shupe_04, the MicroARSCL product and the Polyfit12 in a case study. It should be noted that PEAKO

was trained with a subset of data from the same distribution as this firstly presented case study. This means that PEAKO has somewhat of an advantage over the other three algorithms when comparing on the training data set. Section 4.2 summarizes the testing phase and presents a comparative independent study case, in which PEAKO-found peaks are again compared to peaks detected by the three other algorithms and validated against HSRL retrievals of liquid water droplets. More case studies are presented in the Appendices.

**4.1    Training phase of the PEAKO algorithm**

The training phase was split into two steps: Initially, peaks marked manually in 1340 Doppler spectra were used for training the PEAKO algorithm and obtaining an initial model via a coarse parameter search. This initial training resulted in six equally well performing combinations of span, prominence threshold and width, which all yield the same value of the similarity measure. A more fine-resolved search for the three parameters was then performed, using 775 Doppler spectra with user-marked peaks.

This second, refined training again resulted in several combinations of minimum peak prominence, minimum peak width, and smoothing span yielding the same similarity. Table 2 gives an overview of the possible ranges found for the three PEAKO parameters after the initial training and the finer-resolved parameter search. The span for loess smoothing became slightly larger (increased by 0.5 - 1% in absolute terms) and converged to one single possible value (8.5%). The minimum peak prominence decreased by one third, i.e. from $0.15\,\mathrm{dBZm^{-1}s^{-1}}$ to $0.1\,\mathrm{dBZm^{-1}s^{-1}}$. This prominence threshold is very low compared to

values used by other peak-finding techniques. However, in other approaches, spectra are usually not smoothed and neighbor-averaged. Features prominent enough in time and height to be still visible at all after averaging and smoothing are most probably physical, justifying the low prominence threshold. The possible values for the minimum peak width did not change



**Table 2.** Ranges of the parameters yielding the highest similarity measure after the initial and the fine-tuned training using the first and second training data set, respectively.

| | optimal parameter range after initial training | optimal parameter range after fine-tuned training |
|---|---|---|
| span for smoothing | 7.5 - 8% | 8.5% |
| peak prominence threshold | 0.15 dBZm$^{-1}$s$^{-1}$ | 0.1 dBZm$^{-1}$s$^{-1}$ |
| minimum peak width | 4.2 - 6.3 V$_D$ bins | 4 - 6.25 V$_D$ bins |
| | (9.95 - 14.93 cm s$^{-1}$) | (9.48 - 14.81 cm s$^{-1}$) |

significantly between the initial and the more refined model and ranges between 0.09 to 0.15 m s$^{-1}$ (i.e., V$_D$ range, from 4 to 6.25 V$_D$ bins for the given KAZR Doppler spectra resolution). Doppler spectrum peaks detected by PEAKO configured in one of these combinations (span = 8.5%; prominence treshold =0.1 dBZm$^{-1}$s$^{-1}$; minimum peak width =4 m s$^{-1}$) are compared to peaks found by other methods for a study case on 21 February 2014, 22.54 to 22.77 UTC, at 2 to 6 km height. This parameter set containing the smallest possible minimum peak width was chosen because it is most stringentand thus best suited for the detection of narrow supercooled liquid water peaks. The selected period was discussed in detail in Kalesse et al. (2016).

Fig. 4 shows the first three radar moments, i.e. the radar reflectivity factor Ze, the mean Doppler velocity (MDV) and the Doppler spectrum width $\sigma$ for the first selected case study, which is set from 21 February 2014, 22.54 UTC to 22.77 UTC in 2 to 6 km height. This time period is characterized by the passage of a warm occlusion in Hyytiälä, Finland, shown by the continuously lowering frontal snow cloud base characterized by high Ze. A midlevel mixed-phase cloud was present before the front approached. It can be identified by its supercooled liquid water (SLW) layer near cloud top between approximately 2.9 and 3.2 km altitude. Before the snow cloud moves in (22.54 to 22.69 UTC), new ice is formed from this SLW layer and growing in size while sedimenting, leading to a slight increase in Ze, MDV (absolute value) and $\sigma$ with decreasing altitude. As the frontal cloud moves in and snow begins to fall through the SLW layer, riming takes place along slanted fall streaks. A more in-depth analysis of the synoptic situation and the observed microphysical growth processes is given in Kalesse et al. (2016). The number of detected peaks for this case study is shown in Fig. 5. All algorithms show a similar general picture with an increasing number of spectral peaks as the snow front moves in and snow starts falling through the SLW layer of the midlevel mixed-phase cloud. A closer examination however reveals some differences between the methods: PEAKO and Shupe_04 algorithm have very similar results except for some areas in the snowfall region where PEAKO detects 3 peaks and Shupe_04 detects 2 peaks. MicroARSCL generally shows higher variability than the other algorithms, the small areas of higher peak number often coincide with increased spectrum width in Fig. 4. Polynomial fitting shows 3 peaks in the area where snow falls through the top of the SLW layer and is otherwise very similar to PEAKO and Shupe_04. The areas where the different algorithms show discrepancies are now examined in more detail. For that purpose, contoured frequency by altitude diagrams (CFAD, Fig. 6) are created to compare the results of the algorithms in a different way. The CFAD shows the number of detected peaks (abscissa) at different heights (ordinate) as a colored frequency of occurrence for the total case study sample. For all four compared algorithms, it is most common that only one peak is detected. This is especially true for higher altitudes within the



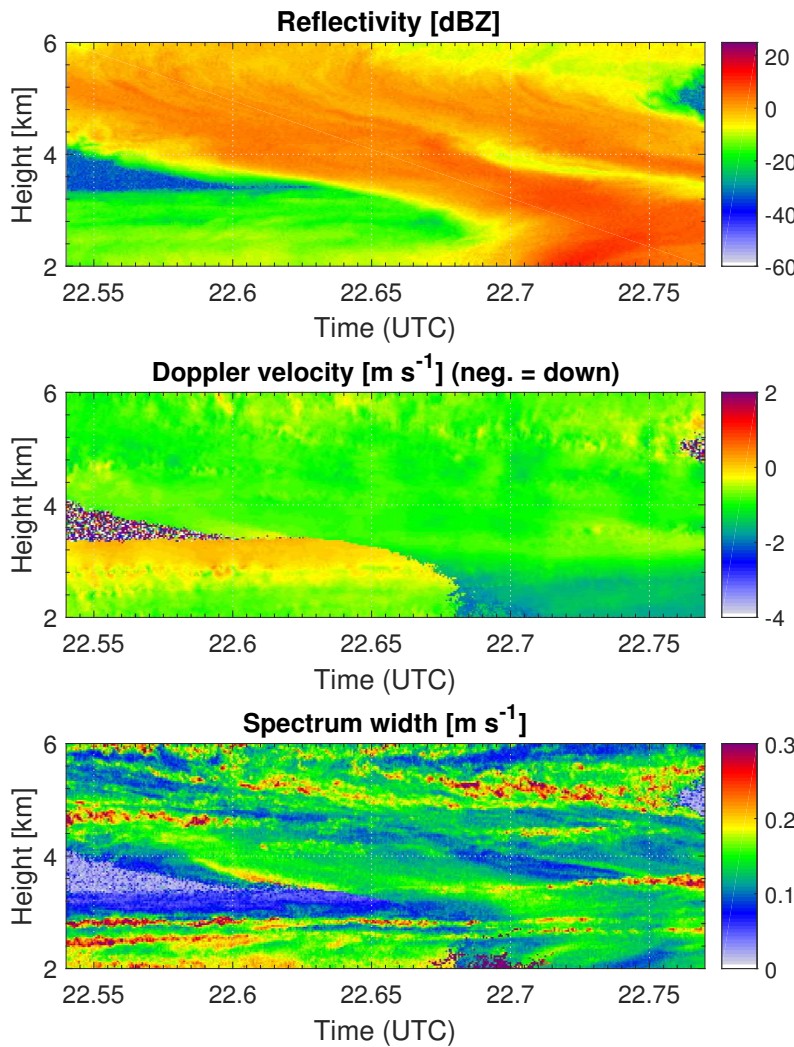

**Figure 4.** Case study period from 2014-02-21 22.54 UTC to 22.77 UTC in 2 to 6 km height. Top to bottom panels show the radar reflectivity factor Ze, the mean Doppler velocity (MDV), and the Doppler spectrum width $\sigma$ of the main peak in the Doppler spectrum.



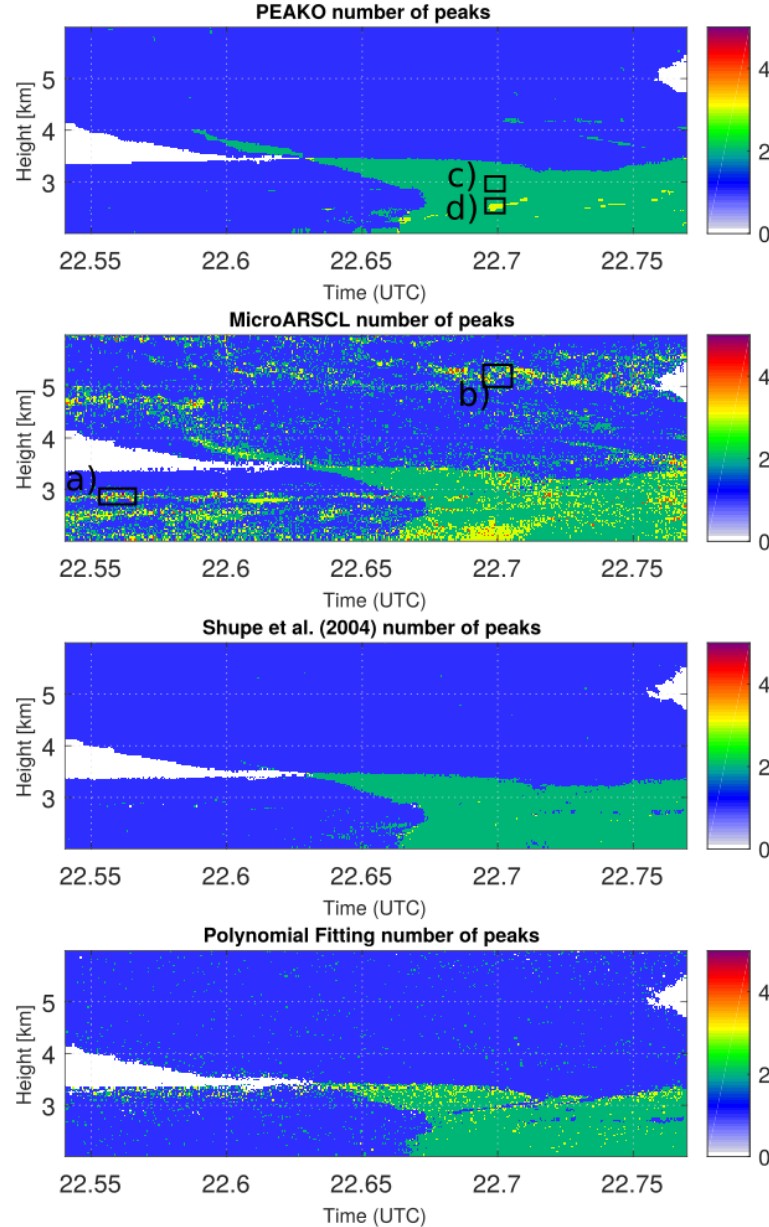

**Figure 5.** Number of Doppler spectrum peaks detected by different algorithms for the selected case study on 2014-02-21 from 22.54 to 22.77 UTC in 2 to 6 km altitude. Top to bottom: Number of peaks found by the PEAKO algorithm for one of the "best parameter" combinations obtained in the training phase of the algorithm; Number of peaks in MicroARSCL data product; Number of peaks detected using the criteria of Shupe_04; Number of peaks determined by Polyfit12





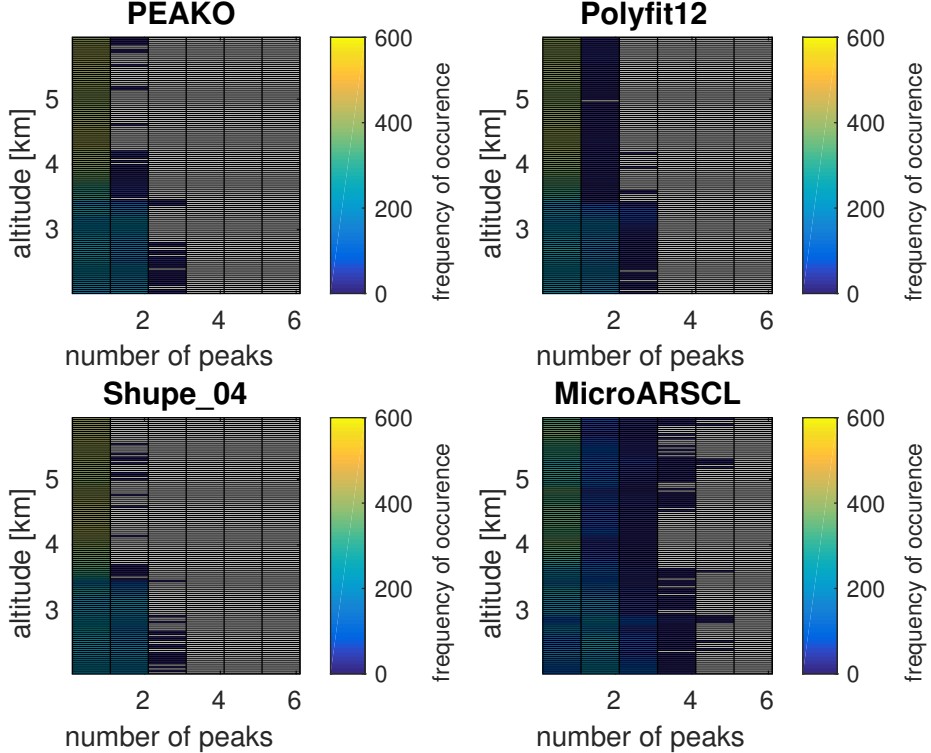

**Figure 6.** Contoured frequency by altitude diagram (CFAD) for the frequency of occurrence of number of detected peaks from different algorithms for the case study period on 2014-02-21 from 22.54 to 22.77 UTC in 2 to 6 km altitude.

snow front (4 km and above). It is visible that the PEAKO algorithm and the results obtained using the Shupe_04 approach agree to a large extent. In the polynomial fitting approach, two or three peaks are detected more often, especially in the layer just above 3 km altitude, where several spectra are classified to contain three peaks. The MicroARSCL data product contains even more Doppler peaks, often three or more, over the complete altitude range.

5  In Fig. 7, four exemplary spectra from regions where the algorithms show discrepancies are shown along with the peaks detected by each of the four algorithms. The spectrum in Fig. 7a is recorded in 2.83 km height at 22.56 UTC, below the SLW and before the snow front moves in. As discussed in Kalesse et al. (2016), ice particles which are nucleated in the SLW layer of the midlevel cloud and growth through water vapor deposition lead to this Gaussian shaped monomodal Doppler peak. The spectrum is relatively broad (as can also be seen from Fig. 4) and noisy, pointing to turbulence. MicroARSCL is sensitive to

10  small-scale noise of the original spectrum which the other algorithms are not sensitive to and thus overestimates the number of spectral peaks. Fig. 7b shows a spectrum from later on, at 22.7 UTC, in the upper part of the frontal snow cloud in 5.26 km height. In this time-height region, PEAKO, Shupe_04 and Polyfit12 detect the Gaussian-shaped snow peak, but MicroARSCL is again sensitive to small-scale fluctuations in the Doppler spectra and finds three peaks. The example in Fig. 7c is taken from



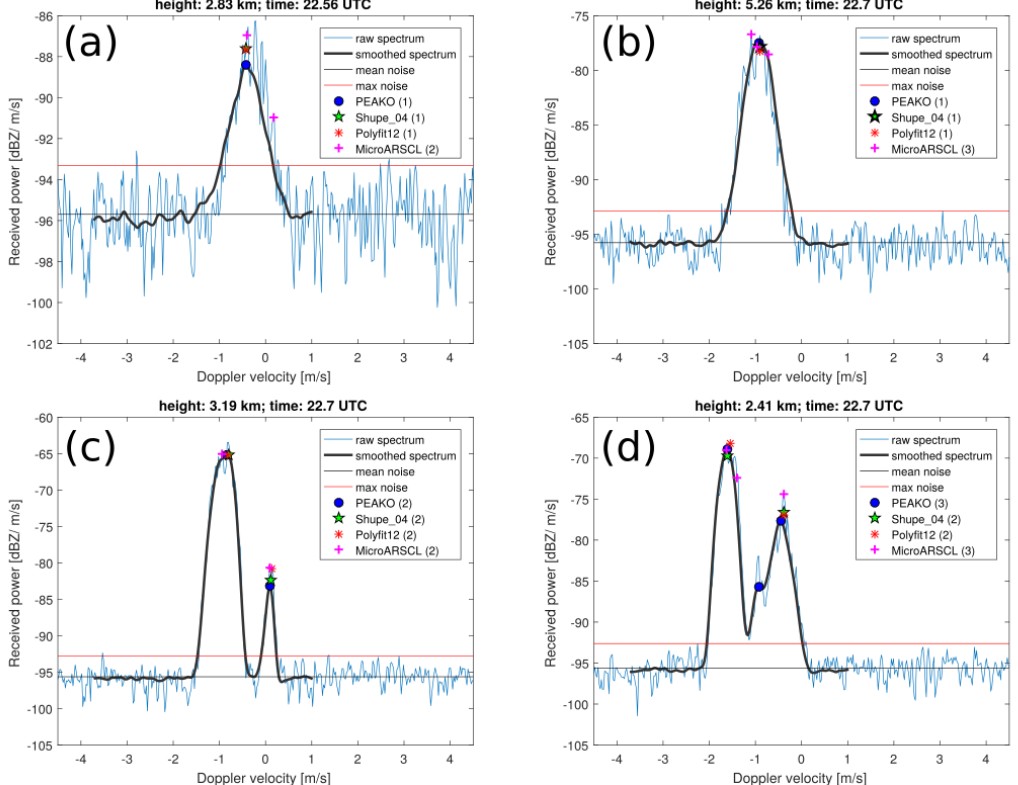

**Figure 7.** Four exemplary Doppler spectra picked from the case study on 21 February 2014, 22.54 to 22.77 UTC in 2 to 6 km altitude. The averaged and smoothed spectrum which is used as input to PEAKO is drawn in bold over the original spectrum. The peaks detected by the four algorithms are marked. The number of peaks found by each algorithm is noted in parenthesis in each figure legend. Please note the different y-scales.

3.2 km altitude at 22.7 UTC where snow from the frontal cloud starts to fall through the SLW. All four algorithms are able to detect the very narrow noise-floor separated peak produced by the supercooled liquid droplets with $V_D$ near $0\,\mathrm{m\,s^{-1}}$. Fig. 7d shows a spectrum recorded at the same time but below the SLW layer in 2.41 km height, where freshly generated ice ($V_D =$ $-0.5\,\mathrm{m\,s^{-1}}$), unrimed snow ($V_D = -1\,\mathrm{m\,s^{-1}}$), and rimed snow (($V_D = -1.7\,\mathrm{m\,s^{-1}}$)) are present. These hydrometeor populations
5   do not have sufficient differences in terminal fall velocities and thus produce a spectrum with merged peaks. For this example, Shupe_04 and Polyfit12 detect the two main maxima (freshly generated ice and rimed snow), whereas MicroARSCL and PEAKO find three peaks. However, closer examination of the example spectrum shows that the two algorithms each detect an additional peak in different locations: MicroARSCL finds a sub-peak in the faster falling hydrometeor population (the Gaussian peak of the rimed snow) while PEAKO detects a sub-peak in the slower-falling mode which exhibits a strong skewness. This
10   subpeak is most likely caused by snow from the upper layers which remained unrimed as discussed in Kalesse et al. (2016).





The results of the PEAKO comparison to the other three peak-finding approaches for the other two training data sets, i.e., for 16 February 2014, 0.67 to 0.92 UTC and for 21 February 2014, 23.01 to 23.25 UTC are shown in Appendix A and B. Time-height plots of peak number are shown in Fig. A2 and B2; CFAD diagrams can be found in Fig. A3 and B3. Both of these time periods were as well analyzed by Kneifel et al. (2015) in depth, so it was possible to compare the microphysical signatures

reported in this study to the Doppler peaks detected by the four algorithms. For 16 February 2014, 0.67 to 0.92 UTC, Kneifel et al. (2015) reported high values of microwave radiometer derived liquid water path of 100 to 500 $g\,m^{-2}$ and clear signatures of large aggregates in the dual-wavelength ratios of Ka-W-band below 2 km as well as in the reflectivity fall streak feature at 0.85 UTC which were also seen in the X-Ka dual-wavelength ratios around 0.85 UTC. The general structure of the layers of Doppler peak number detected by PEAKO, Shupe_04 and Polyfit12 again agree to a large extent, whereas MicroARSCL

detects a higher number of peaks in both cases (Fig. A2 and B2). The fall streak feature which exhibits particle size sorting was better detected by PEAKO and MicroARSCL than by the other two algorithms. The increase in number of peaks at 0.67 to 0.75 UTC can be explained by the presence of large needle aggregates of sometimes more compact and sometimes very open structure as explained in (Kneifel et al., 2015) which lead to the interesting multi-modal Doppler spectrum with up to four peaks as shown in Fig. B4d). Ground-based in-situ observations show that during 0.75 - 0.85 UTC aggregates and rimed particles

with enhanced terminal velocities were present and that the number of large aggregates was further found to decrease while number of increasingly rimed aggregates further increased until 1 UTC. Radio sounding observations on 15 February 2014 at 23.2 UTC show a thin layer at 0.8-0.9 km altitude which is subsaturated with respect to ice and liquid and which might explain the decrease to 1 found Doppler peak at this altitude. For 21 February 2014, 23.01 to 23.25 UTC, Kneifel et al. (2015) report the transition from a low concentration of strongly rimed particles (lump graupel) to aggregate snowfall with large snowfall

rates and increasing size and number of the aggregates. The fast transition of the snowfall from rimed particles to aggregates results in the bimodal Doppler spectra (with two found peaks) at 23 to 23.05 UTC and monomodal spectra afterwards. For this case study, PEAKO and Shupe_04 and Polyfit12 agree well with the situation described in Kneifel et al. (2015) while MicroARSCL overestimates the number of peaks especially in the turbulent boundary layer and near 4 km altitude.

## 4.2 Testing phase of the algorithm

Using the tuned parameter pairs obtained in the training phase, the PEAKO algorithm is again compared to the other three algorithms, as well as to data measured by an independent instrument, the HSRL. For this purpose, a case of a frontal passage associated with snow on 02 February 2014, 16 to 17 UTC was analyzed. During this time, a liquid-topped mixed-phase cloud with cloud top temperature of (T = -4°C) and cloud top height of 2.6 km was present (Fig. 9). A deeper precipitating cloud system with cloud top around 8 km (cloud top temperature of -40°C) was approaching the TMP site at about 16.27 UTC. The

surface temperature was -5°C. During the first half of the hour-long case study, the HSRL detected an embedded layer of SLW in the mid-level cloud, characterized by high backscatter coefficient values and low depolarisation ratio values (Luke et al., 2010) in Fig. 8. The SLW layer is located between 0.8 and 1 km height and is slightly lifted as the front is moving in. Its base and top are traced with dashed lines in Fig. 8. After 16.8 UTC, the microwave radiometer derived LWP decreases from around 300 $g\,m^2$ to approximately 60 $g\,m^2$ and the lidar does not detect the cloud base anymore due to the scarcity in small liquid



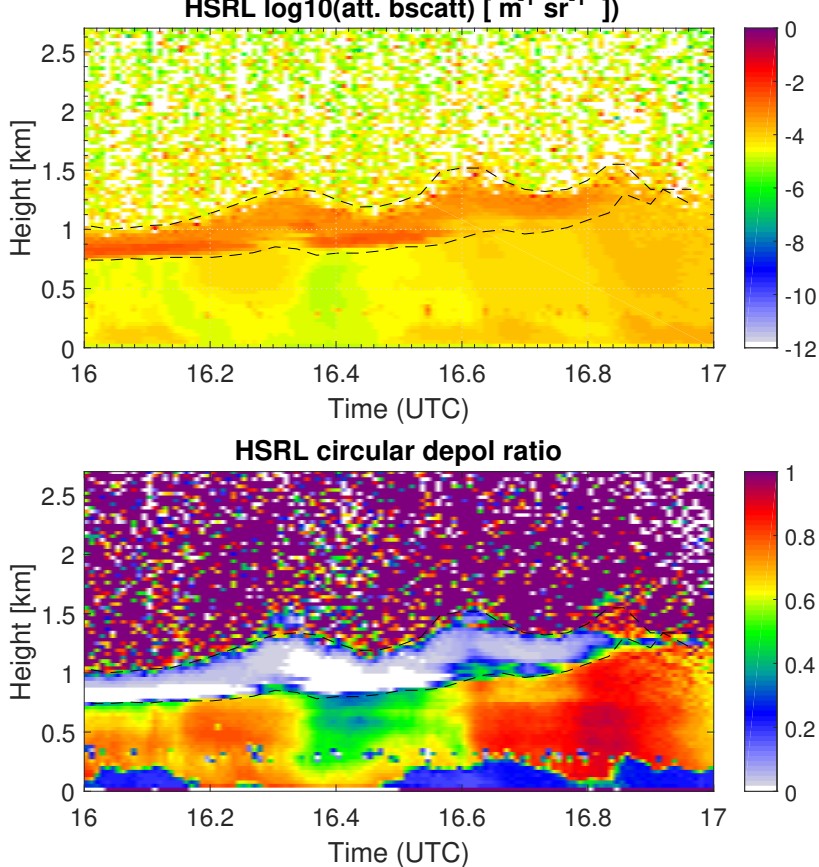

**Figure 8.** HSRL measurements from the case study period from 02 February 16 UTC to 17 UTC in 0 to 3 km height. The top panel shows the attenuated backscatter cross section and the lower panel the circular depolarisation ratio.

droplets to which the lidar is sensitive to and due to the strong snow fall. The strong decrease in LWP is again pointing to riming, which is substantiated by a closer look at the Doppler spectra (Fig. 12). Analysis of the ground-based in-situ Particle Imaging Package (PIP) data shows a variety of different precipitating particles during that one hour time period (Annakaisa von Lerber, personal communication): Around around 16 UTC, oblate particles, possibly needles, and some small needle aggregates are present. When Ze decreases around 16.3 - 16.55 UTC (Fig. 9), no large particles are present at all, just very small ones, maybe single pristine crystals (the resolution of PIP is not good enough to distinguish). At 16.55 - 16.8 UTC when Ze increases strongly and LWP decreases significantly, the PIP observes a clear change to round, dense, fast falling particles, indicative of small graupel. Finally, from 16.8 UTC onward, particle sizes at the ground increase, there are more (quite dense) aggregates, resulting in Ze of up to 10 dBZ.





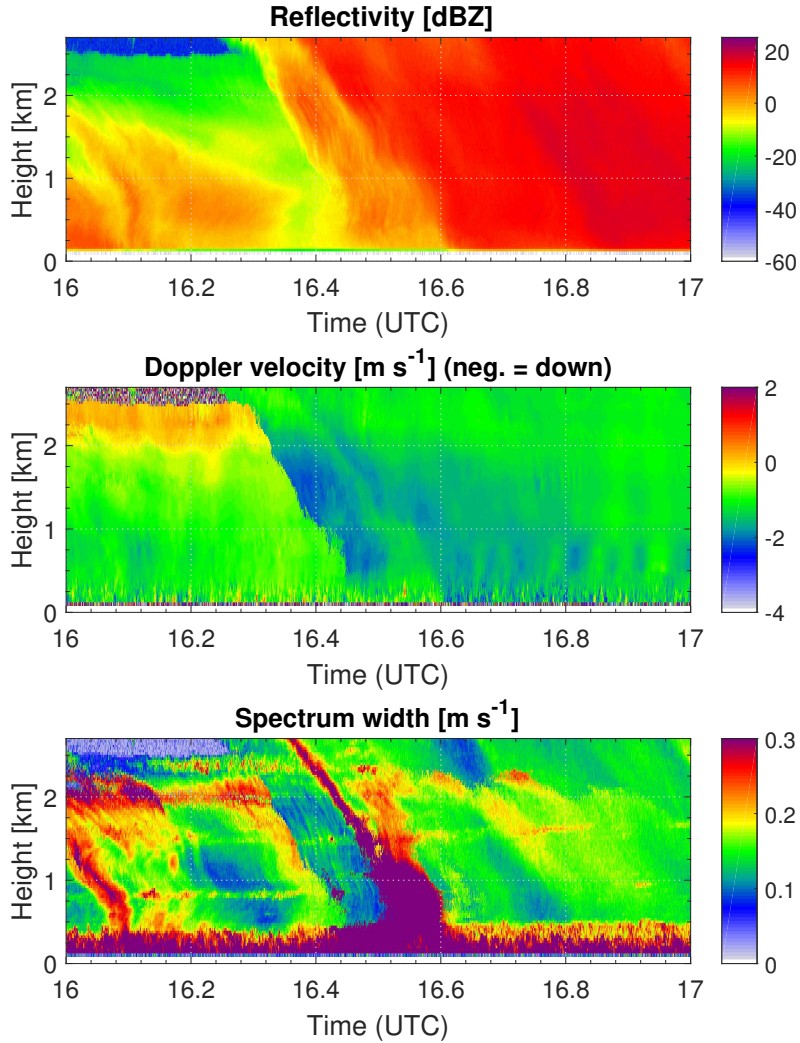

**Figure 9.** Like Fig. 4 but for 02 February 2014, 16.00 to 17.00 UTC in 0 to 2.7 km height.

Fig. 9 shows the first three radar moments (Ze, MDV and $\sigma$) of the main peak for the selected case study. The supercooled liquid layer at the top of the mid-level cloud extends from about 2.1-2.4 km and is characterized by MDV of near $0\,\mathrm{m\,s^{-1}}$. Snow fall rate is at first low and increases at about 16.6 UTC (surface meteorological observations, not shown). Pronounced fall streaks can be seen coinciding with large values of spectrum width, indicating the presence of several hydrometeor populations, producing Doppler spectra with broad merged peaks. Fig. 10 reveals that the number of peaks detected by the four algorithms differ significantly for this case study. Shupe_04 and Polyfit12 again agree to large extents although Shupe_04 does mostly



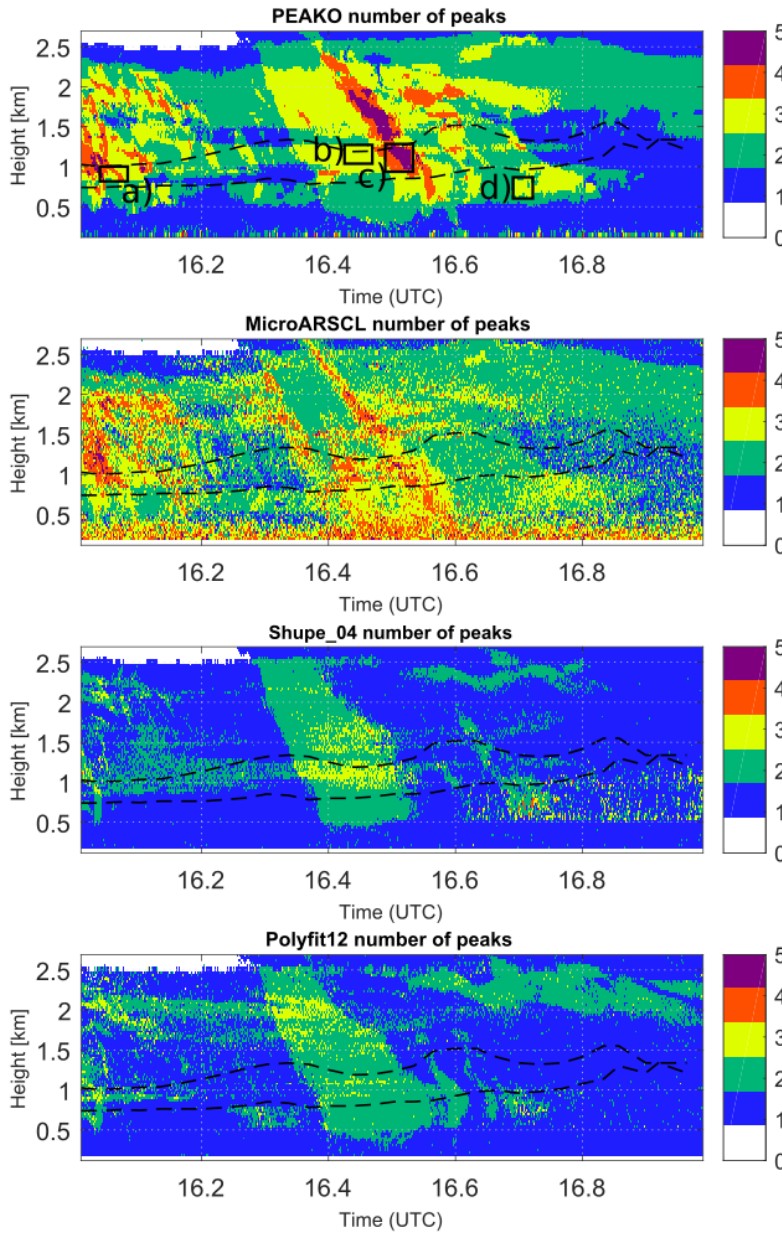

**Figure 10.** Like Fig. 5 but for 02 February 2014, 16 UTC to 17 UTC in 0 to 2.7 km height. Boxes mark the points in time-height where peaks detected in single spectra are analyzed in greater detail. The dashed lines mark the bottom and top of the SLW layer detected by the HSRL (Fig. 8)





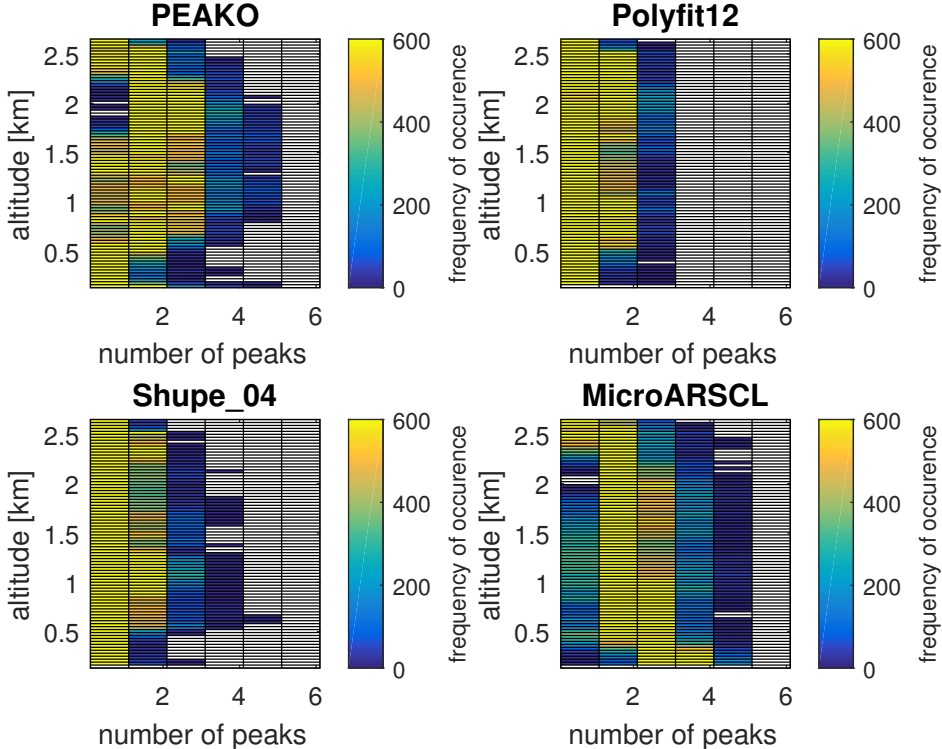

**Figure 11.** Like Fig. 6 but for the case study period on 02 February 2014, 16.00 to 17.00 UTC in 0 to 2.7 km altitude.

only detect one peak at 2-2.5 km height before 16.3 UTC and after 16.5 UTC where all other algorithms mostly detect two peaks. MicroARSCL generally detects a larger number of peaks in the Doppler spectra, PEAKO is in this case more similar to MicroARSCL than to the other two algorithms and often detects three to four and sometimes even five peaks along certain fall streaks. However, due to the smoothing performed within PEAKO, the detected features are less noisy and more consistent in

5   time-and-height than for MicroARSCL.

Fig. 11 shows CFAD diagrams for each of the four algorithms for the case study on 02 February 2014, 16 to 17 UTC. These graphs confirm that Polyfit12 and Shupe_04 estimate the number of peaks more conservatively than PEAKO and MicroARSCL. It is obvious that MicroARSCL often detects three or four peaks in the lowermost radar height bins, which can probably be attributed to turbulence. The HSRL-detected SLW layer (varying between 0.7 to 1.5 km height) is most obvious in the Shupe_04

10   and PEAKO CFAD plot, the number of spectra which are assigned two or three peaks is noticeably higher in this altitude range.

Fig. 12 shows four exemplary Doppler spectra from the case study on 02 February 2014, 16 to 17 UTC. Only PEAKO is able to detect a narrow peak near $0\,\mathrm{m\,s^{-1}}$ in all four example spectra. While these peaks can be attributed to SLW in Fig. 12a, Fig. 12b, and Fig. 12c, it is more likely that this small subpeak in Fig. 12d is caused by small ice ice particles nucleated in the SLW layer situated slightly above because the HSRL does not detect liquid at 0.7 km altitude around 16.7 UTC. Besides




the peak near $0\,\mathrm{m\,s^{-1}}$, all shown spectra are characterized by broad merged snow peaks pointing to snow particles of different size, shape, and density falling at different terminal velocities. In Fig. 12a, the three merged modes of snow, as well as the SLW peak are detected by PEAKO and MicroARSCL, while Shupe_04 and Polyfit12 both only detect one maximum. The spectrum shown in Fig. 12b is near the top of the SLW layer detected by the HSRL. The narrow liquid peak with fall velocity

near $0\,\mathrm{m\,s^{-1}}$ is only detected by PEAKO and Shupe_04. Both algorithms find two more snow peaks with larger fall velocity. These two peaks are also detected by Polyfit12. MicroARSCL detects three peaks as well, however is not able to detect the liquid peak. Fig. 12c shows a spectrum which was chosen in an area where PEAKO finds five peaks. Again, one of them is a SLW peak within the SLW layer detected by the HSRL Fig. 10. This peak is also detected by MicroARSCL and Shupe_04 but not Polyfit12. The four other peaks found by PEAKO are merged snow peaks with different fall velocities which hint to various

degrees of riming that the other algorithms have difficulties to detect. In Fig. 12d, the number of peaks detected by the four algorithms differ significantly: The peak with highest reflectivity at around $-1.5\,\mathrm{m\,s^{-1}}$ fall velocity is found by all algorithms. PEAKO detects two sub-peaks, which are each detected by at least one other algorithm as well. However, none of the other methods finds both other ice sub-peaks.

In the Appendices A-C three more case studies from the training and test phase are presented. Comparative results of PEAKO

to the other peak finding algorithms are similar to the cases presented here.

## 5    Conclusions and Outlook

### 5.1    Summary of findings and outlook

The presented study focuses on the description of a new supervised cloud radar Doppler velocity spectrum peak-finding algorithm (PEAKO). Its performance was compared to different existing Doppler spectrum peak-finding algorithms. It was found

that the PEAKO algorithm generally agrees well with results from Shupe_04 and a polynomial fitting approach. PEAKO is however capable to detect narrower merged peaks with a smaller power contribution than Shupe_04. The polynomial fitting approach has mostly similar results as Shupe_04 but is not very practical due to its long computation time. The MicroARSCL product was usually more sensitive to small perturbations in the radar Doppler spectrum and thus often detected a higher number of peaks than the other three algorithms and produces more "speckled" results. Some areas where peaks are overestimated

by MicroARSCL are in high-turbulent regions with large spectrum width like the turbulent boundary layer while others seem more random and not consistent in time and height. Consistency in time and to a lesser extent height is a good indicator of the performance of a peak-finding algorithm because hydrometeor populations and cloud microphysical processes generally occur in layers (unless in high turbulent regions). The number of found cloud radar Doppler velocity spectrum peaks within mixed-phase wintertime snow clouds in Finland where validated with independent ground-based in-situ observations described

in Kneifel et al. (2015) and if available HSRL observations. In upcoming projects, it is planned to test if the found best three-parameter pairs of PEAKO can easily be applied to other radar systems (like METEK-MIRA 35GHz radars or RPG 94GHz FMCW radars) or to which extent further refinement is needed for different radar sampling parameters. Additionally, the effect of stronger cloud dynamics will be evaluated. Determining the *number* of different hydrometeor populations in the same radar





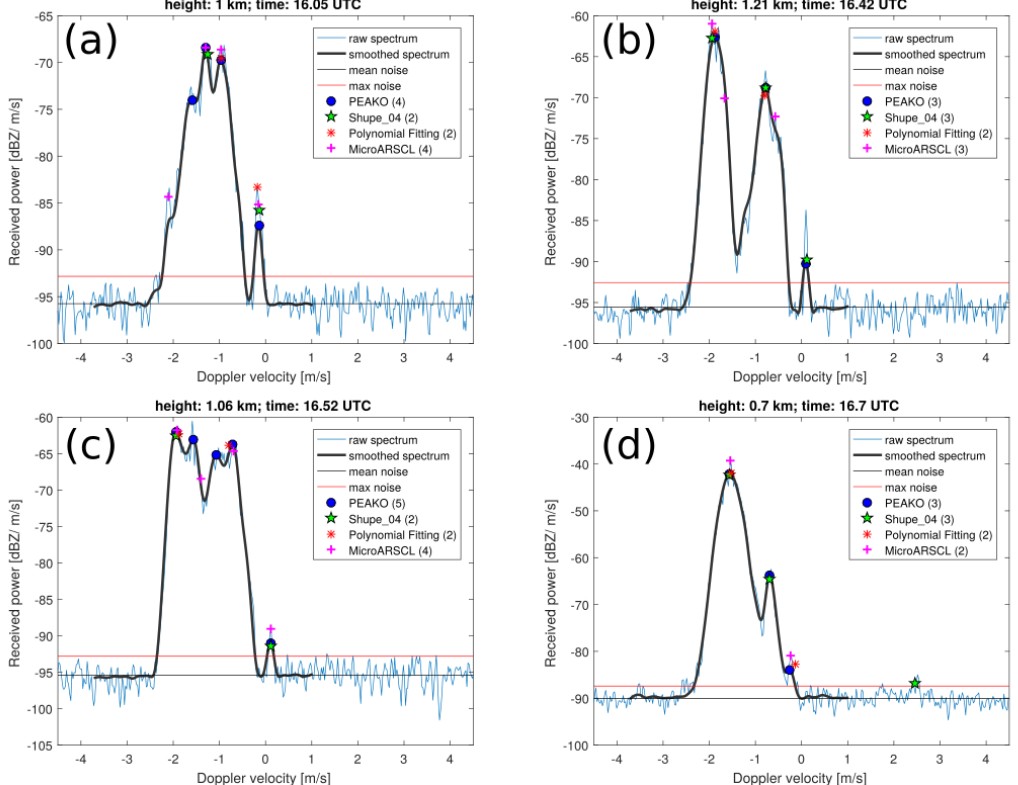

**Figure 12.** Four exemplary Doppler spectra picked from the case study on 02 February 2014, 16 to 17 UTC in 0 to 2.5 km altitude. The averaged and smoothed spectrum which is used as input to PEAKO is drawn in bold over the original spectrum. The peaks detected by the four algorithms are marked. The number of peaks found by each algorithm is noted in parenthesis in each figure legend. Mean and maximum noise floor are presented by black and red horizontal lines, respectively. Please note that the scale of the y axis is different in each plot.

volume based on morphological features of the radar Doppler spectrum as presented in this comparative study is the first step towards cloud particle classifications which is however not the focus of this paper.

*Data availability.* All KAZR and HSRL data used in this study are publicly accessible at the ARM data archive: www.archive.arm.gov.

Atmospheric Radiation Measurement (ARM) user facility. 2014, updated hourly. High Spectral Resolution Lidar (HSRL). 2014-02-02

5   to 2014-02-21, ARM Mobile Facility (TMP) U. of Helsinki Research Station (SMEAR II), Hyytiala, Finland; AMF2 (M1). Compiled by B. Ermold, E. Eloranta, H. Michelsen, J. Garcia, J. Goldsmith and R. Bambha. ARM Data Center. Data set accessed 2018-11-27 at http://dx.doi.org/10.5439/1025200.

Atmospheric Radiation Measurement (ARM) user facility. 2014, updated hourly. Ka ARM Zenith Radar (KAZRGE). 2014-02-02 to 2014-02-21, ARM Mobile Facility (TMP) U. of Helsinki Research Station (SMEAR II), Hyytiala, Finland; AMF2 (M1). Compiled by A.

10  Matthews, B. Isom, D. Nelson, I. Lindenmaier, J. Hardin, K. Johnson and N. Bharadwaj. ARM Data Center. Data set accessed 2018-11-



27 at http://dx.doi.org/10.5439/1025214.

Atmospheric Radiation Measurement (ARM) user facility. 2014, updated hourly. Ka ARM Zenith Radar (KAZRSPECCMASKGECOPOL). 2014-02-02 to 2014-02-21, ARM Mobile Facility (TMP) U. of Helsinki Research Station (SMEAR II), Hyytiala, Finland; AMF2 (M1). Compiled by A. Matthews, B. Isom, D. Nelson, I. Lindenmaier, J. Hardin, K. Johnson and N. Bharadwaj. ARM Data Center. Data set
5  accessed 2018-11-27 at http://dx.doi.org/10.5439/1025218.



**Appendix A:  Case study from 16 February 2014, 0.67 - 0.92 UTC (training data set 2)**

The results of the PEAKO comparison to the other three peak-finding approaches for the training data set of 16 February 2014, 0.67 to 0.92 UTC is shown in Appendix. This time period is also analyzed by Kneifel et al. (2015) in depth, so it was possible to compare the microphysical signatures reported in this study to the Doppler peaks detected by the four algorithms. For 16 February 2014, 0.67 to 0.92 UTC, Kneifel et al. (2015) reported high values of microwave radiometer derived liquid water path of 100 to 500 $\mathrm{g\,m^{-2}}$ and clear signatures of large aggregates in the dual-wavelength ratios of Ka-W-band below 2 km as well as in the reflectivity fall streak feature at 0.85 UTC which were also seen in the X-Ka dual-wavelength ratios around 0.85 UTC. The general structure of the layers of Doppler peak number detected by PEAKO, Shupe_04 and Polyfit12 again agree to a large extent, whereas MicroARSCL detects a higher number of peaks in both cases (Fig. A2 and B2). The fall streak feature which exhibits particle size sorting was better detected by PEAKO and MicroARSCL than by the other two algorithms. The increase in number of peaks at 0.67 to 0.75 UTC can be explained by the presence of large needle aggregates of sometimes more compact and sometimes very open structure as explained in (Kneifel et al., 2015) which lead to the interesting multi-modal Doppler spectrum with up to four peaks as shown in Fig. B4d). Ground-based in-situ observations show that during 0.75 - 0.85 UTC aggregates and rimed particles with enhanced terminal velocities were present and that the number of large aggregates was further found to decrease while number of increasingly rimed aggregates further increased until 1 UTC. Radio sounding observations on 15 February 2014 at 23.2 UTC show a thin layer at 0.8-0.9 km altitude which is subsaturated with respect to ice and liquid and which might explain the decrease to 1 found Doppler peak at this altitude.

**Appendix B:  Case study from 21 February 2014, 23.01 - 23.10 UTC (training data set 3)**

The training data set of 21 February 2014, 23.01 to 23.25 UTC which is also a case study of Kneifel et al. (2015) is described in Appendix B. For 23.01 to 23.25 UTC, Kneifel et al. (2015) report the transition from a low concentration of strongly rimed particles (lump graupel) to aggregate snowfall with large snowfall rates and increasing size and number of the aggregates. The fast transition of the snowfall from rimed particles to aggregates results in the bimodal Doppler spectra (with two found peaks) at 23 to 23.05 UTC and monomodal spectra afterwards. For this case study, PEAKO and Shupe_04 and Polyfit12 agree well with the situation described in Kneifel et al. (2015) while MicroARSCL overestimates the number of peaks especially in the turbulent boundary layer and near 4 km altitude.

**Appendix C:  Case study from 7 February 2014, 23.75 - 24 UTC (test data set 2)**

The second test data set of February 7, 2014 23.75 to 24 UTC is characterized by dendritic ice particles (Kneifel et al., 2015) and a slanted fall streak feature extending from near 4 km to 1 km from 23.75 to 23.9 UTC (Fig. C1) with bimodal Doppler spectra (Fig. C2 and Fig. C4). Ground-based in-situ observations report mostly small, open-structured aggregates (which are later replaced by more compact spheroidal habits) as well as a small number of spherical probably rimed particles.



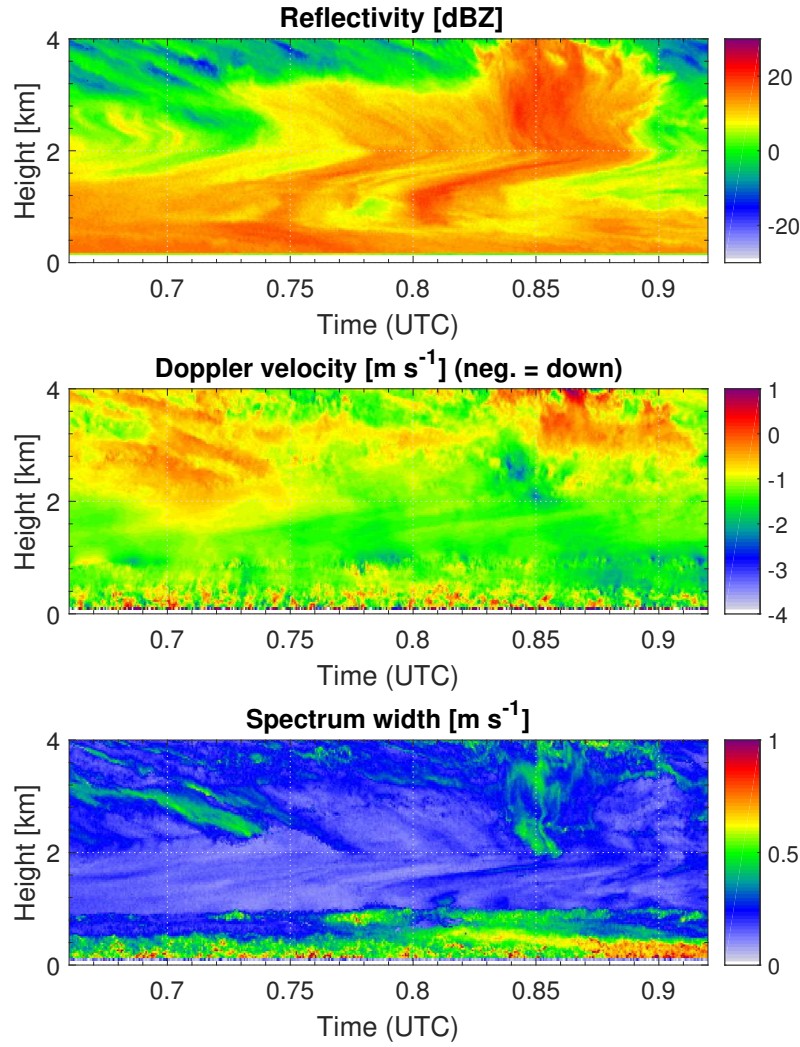

**Figure A1.** Like Fig. 4 but for 16 February 2014, 0.67 - 0.92 UTC in 0 - 4 km height.




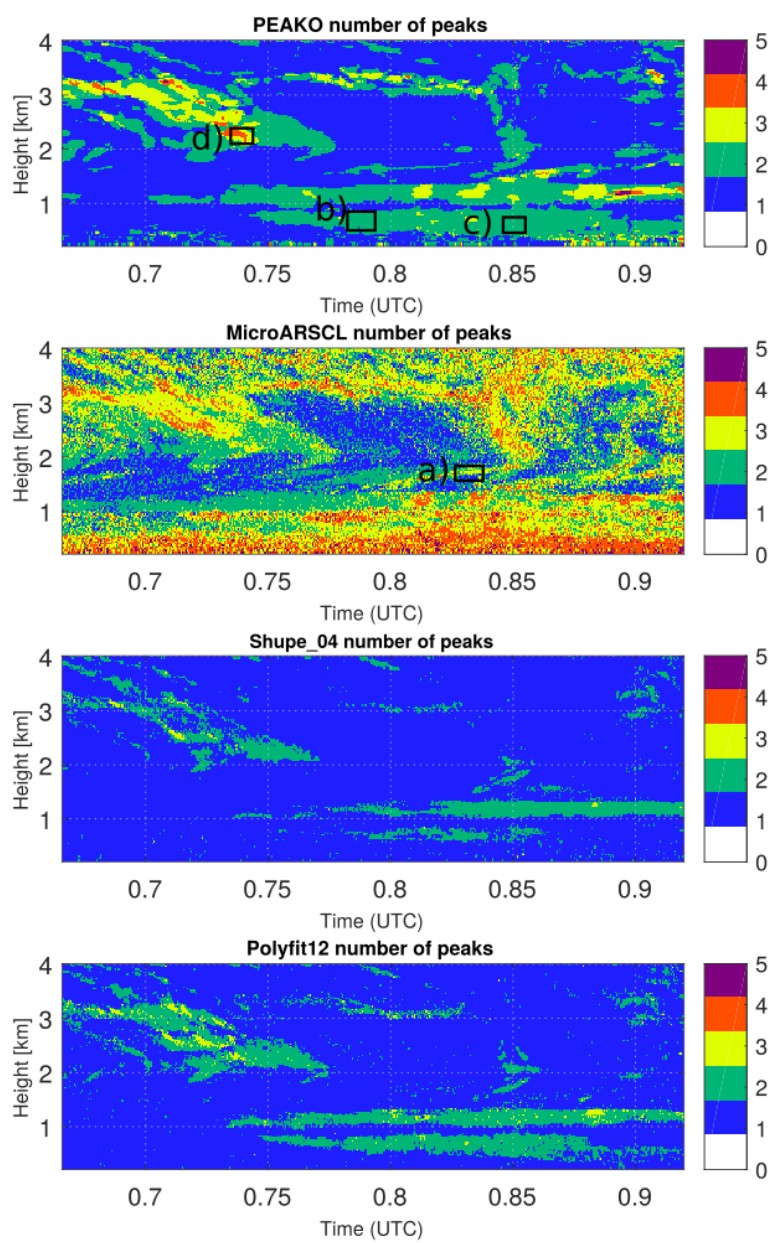

**Figure A2.** Like Fig. 5 but for 16 February 2014, 0.67 - 0.92 UTC in 0 - 4 km height.





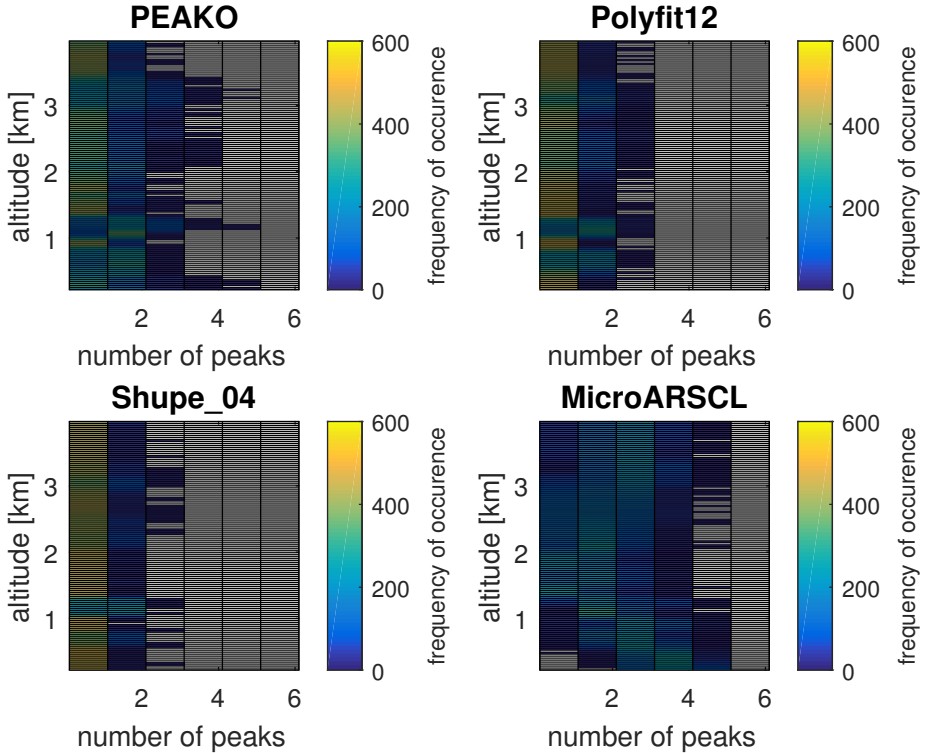

**Figure A3.** Like Fig. 6 but for the case study period on 16 February 2014, 0.66 - 0.92 UTC in 0 - 4 km altitude.

*Author contributions.* Heike Kalesse, Teresa Vogl, and Cosmin Paduraru desgined the PEAKO algorithm together. Cosmin Paduraru developed most of the Matlab model code. Heike Kalesse and Teresa Vogl did the data analysis, visualization, and prepared the manuscript. Edward Luke contributed MicroARSCL data for the selected case studies.

*Competing interests.* The authors declare that they have no conflict of interest.

5  *Acknowledgements.* The authors thank the entire BAECC-SNEX science team, the AMF2 team, and the SMEAR II staff for data acquisition and analysis. Thanks also to Annakaisa von Lerber of the Finnish Meteorological Institute for discussions on ground-based in-situ data as well as Pavlos Kollias for discussions on peak-finding in cloud radar Doppler velocity spectra. Thanks to the Leibniz Institute for Tropospheric Research through which the student research position of Teresa Vogl has received funding from the European Union's Horizon 2020 research and innovation programme under grant agreement No 654109. Heike Kalesse conducted most of this work within the framework of the DFG
10  project COMPoSE, GZ: KA 4162/1-1.



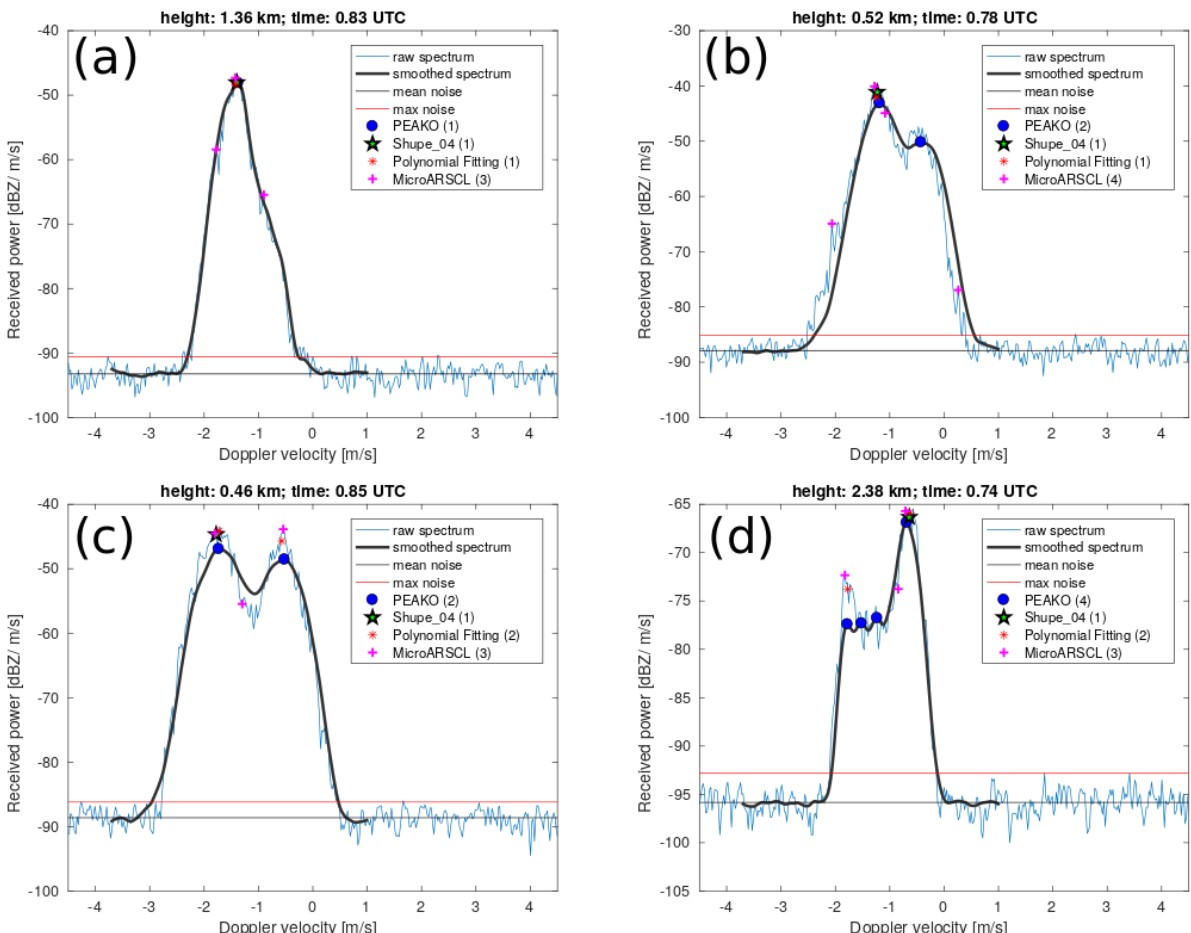

**Figure A4.** Four example spectra selected from the case study on 16 February 2014, 0.63 - 0.92 UTC. Please note that the y axis scale is different for each of the spectrum plots.





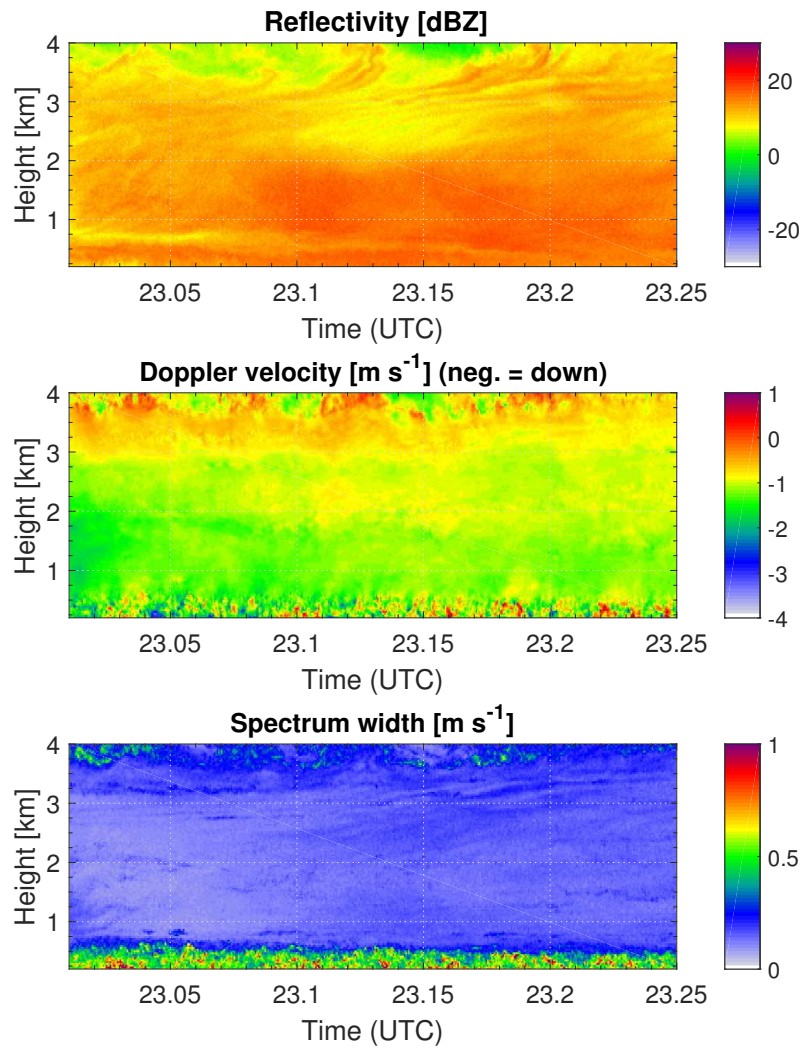

**Figure B1.** Like Fig. 4 but for 21 February 2014, 23.01 - 23.25 UTC in 0.2 - 4 km height.





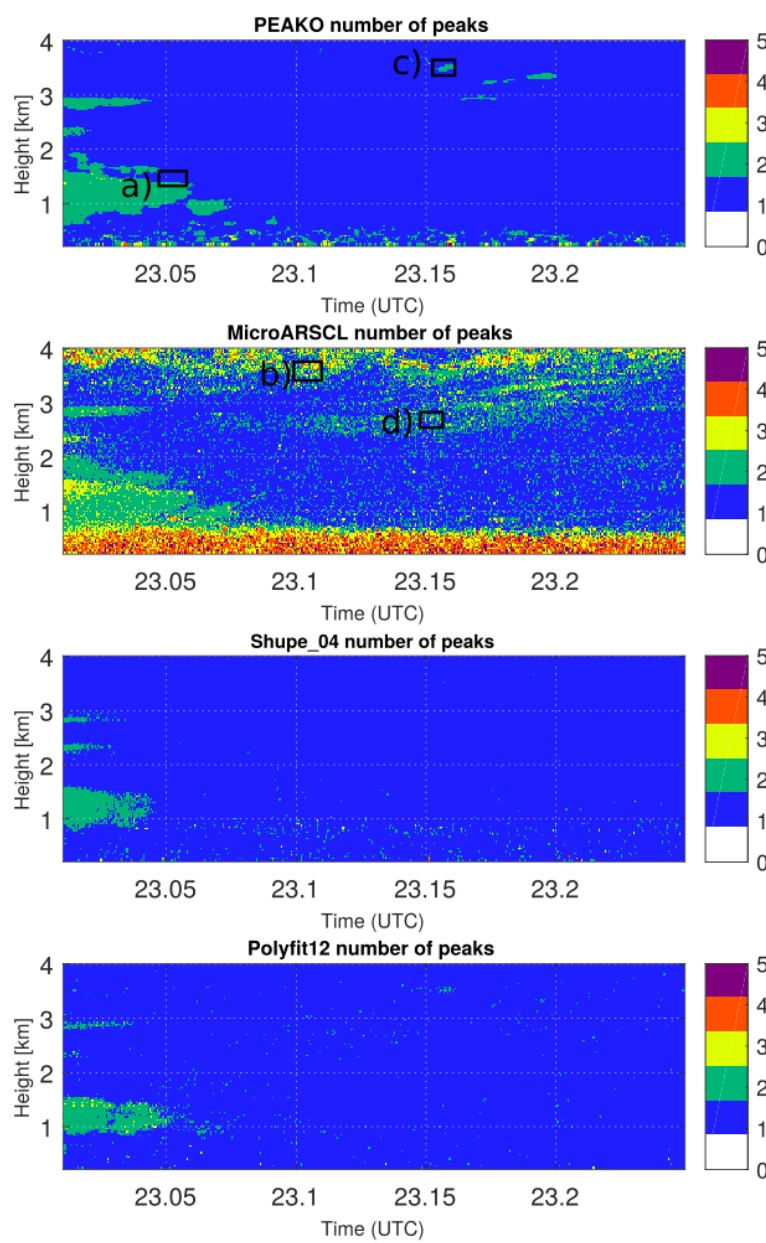

**Figure B2.** Like Fig. 5 but for 21 February 2014, 23.01 - 23.25 UTC in 0.2 - 4 km height.





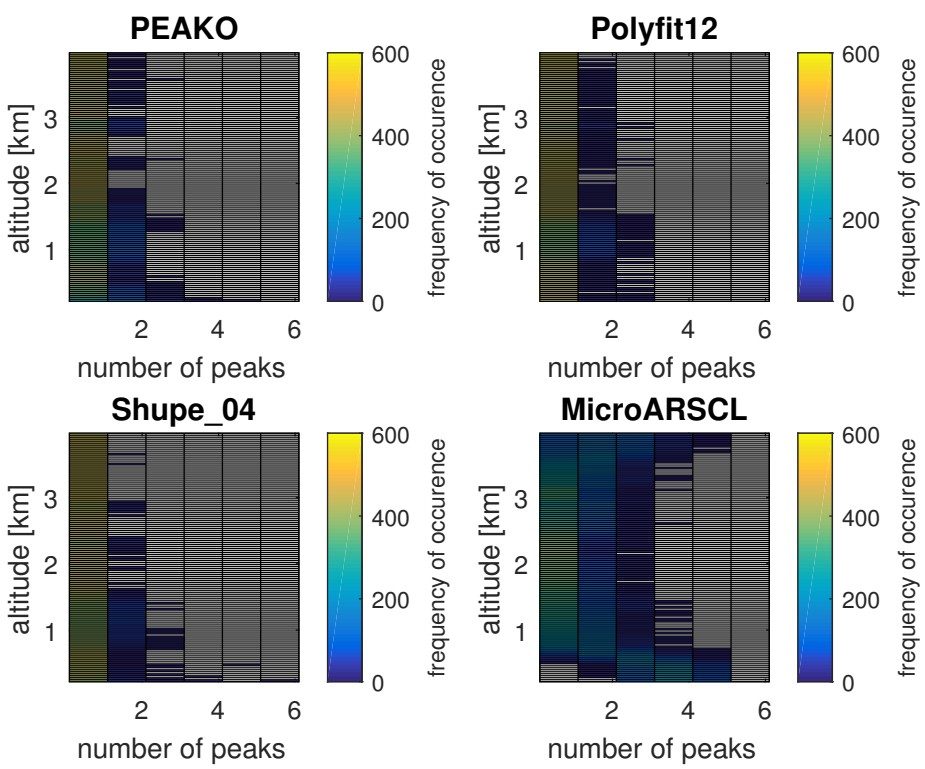

**Figure B3.** Like Fig. 6 but for the case study period on 21 February 2014, 23.01 - 23.25 UTC in 0.2 - 4 km altitude.



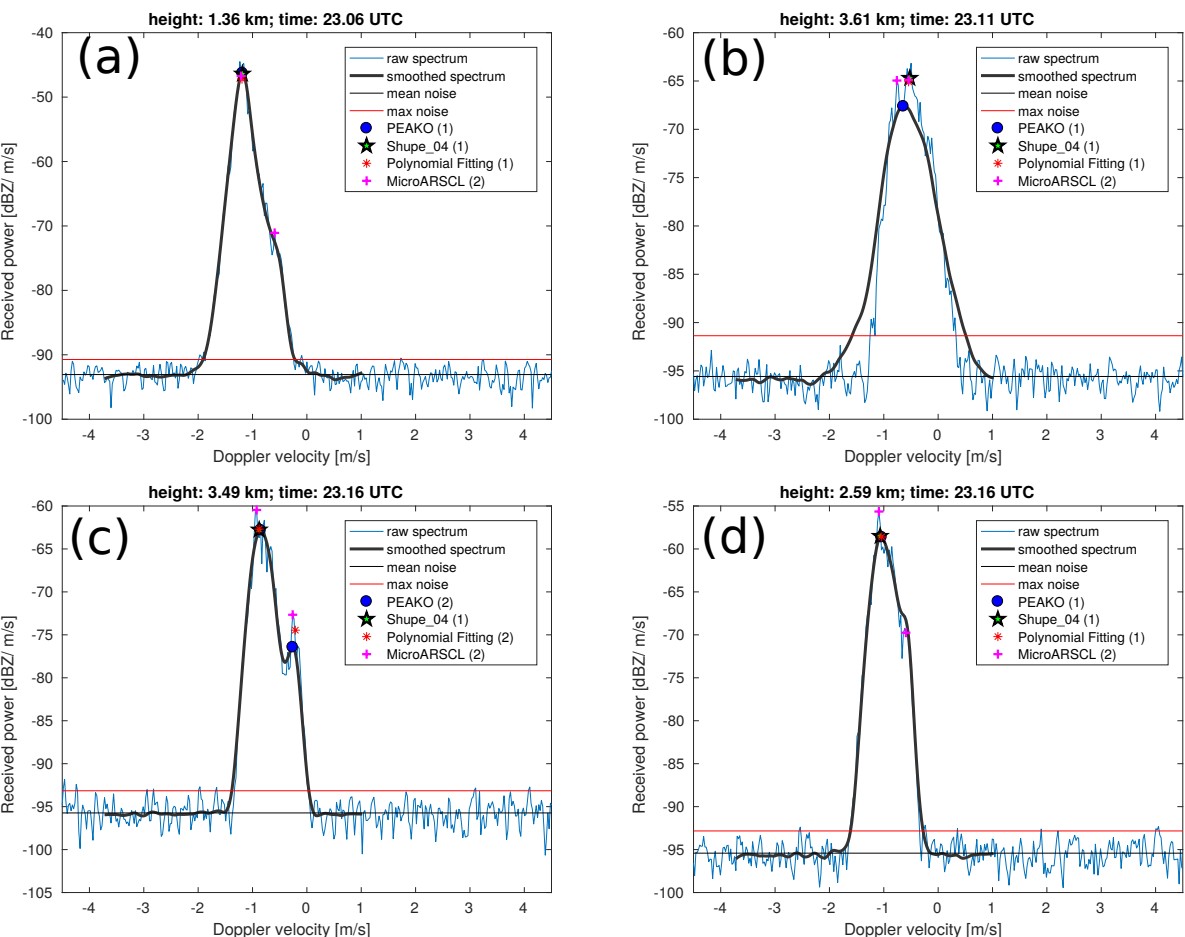

**Figure B4.** Three example spectra selected from the case study on 21 February 2014, 23.01 - 23.25 UTC. Please note that the y axis scale is different for each of the spectrum plots.





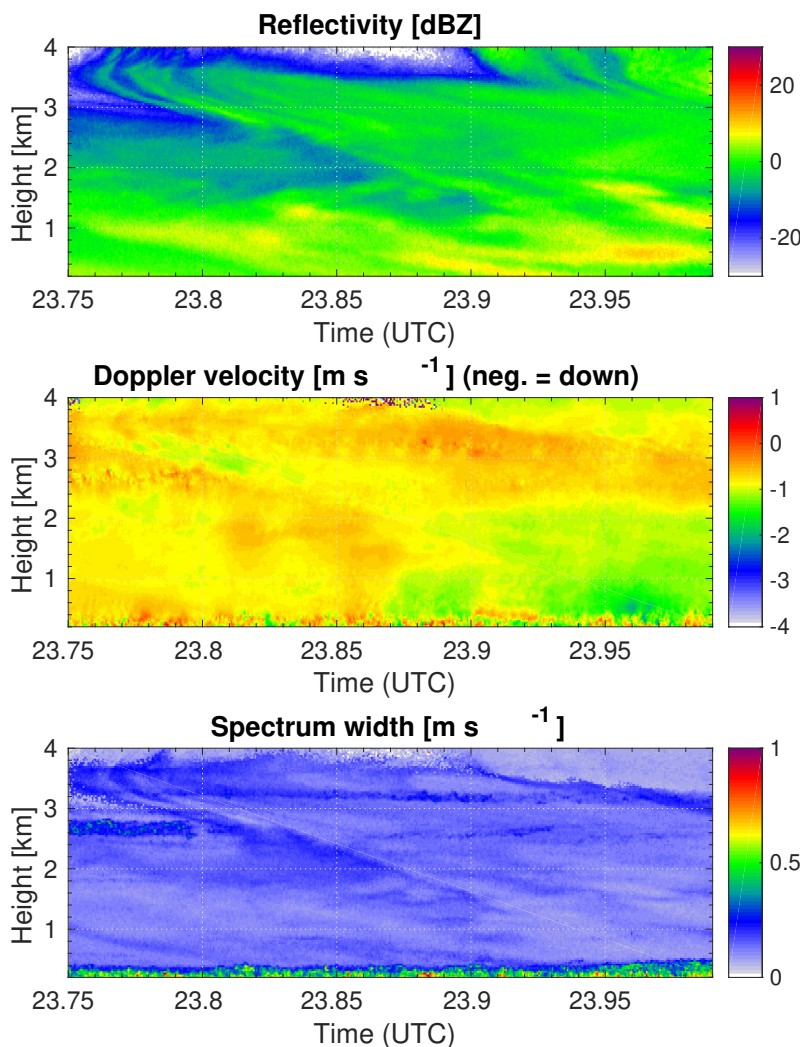

**Figure C1.** Like Fig. 4 but for 7 February 2014, 23.75 - 24 UTC in 0.2 - 4 km height.



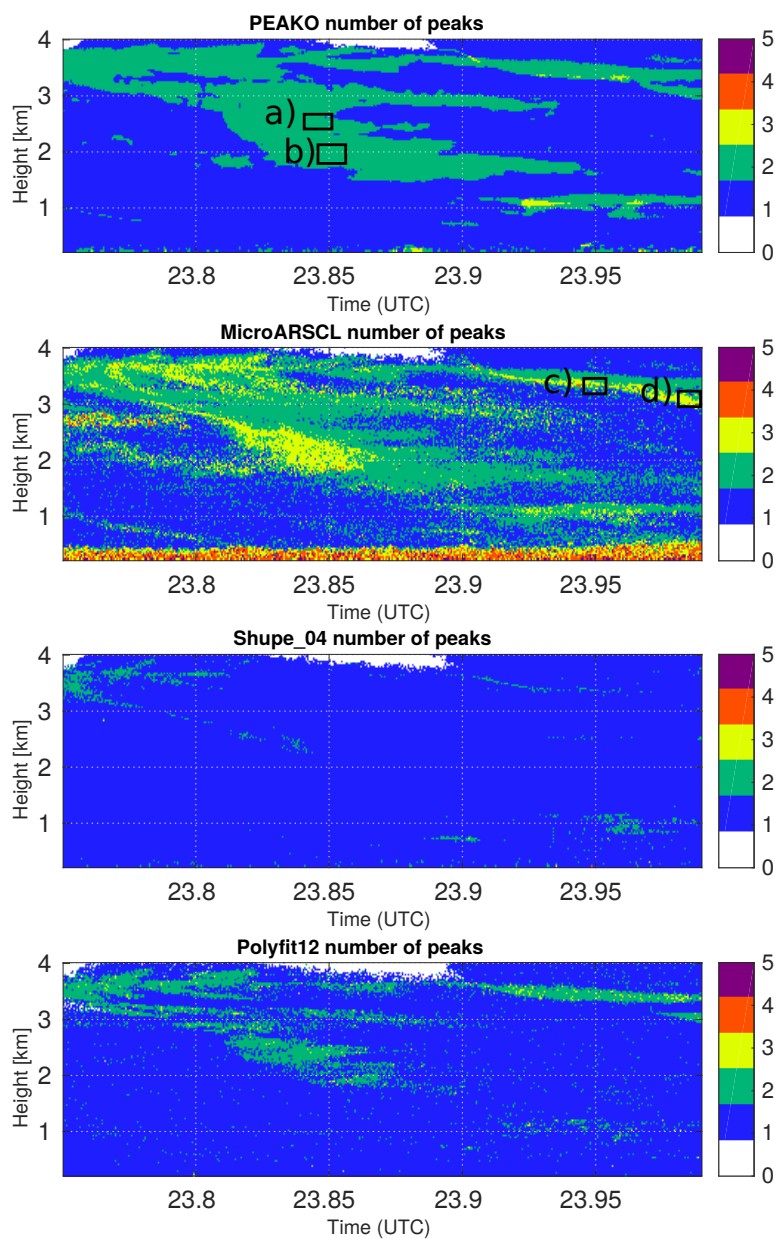

**Figure C2.** Like Fig. 5 but for 7 February 2014, 23.75 - 24 UTC in 0.2 - 4 km height.





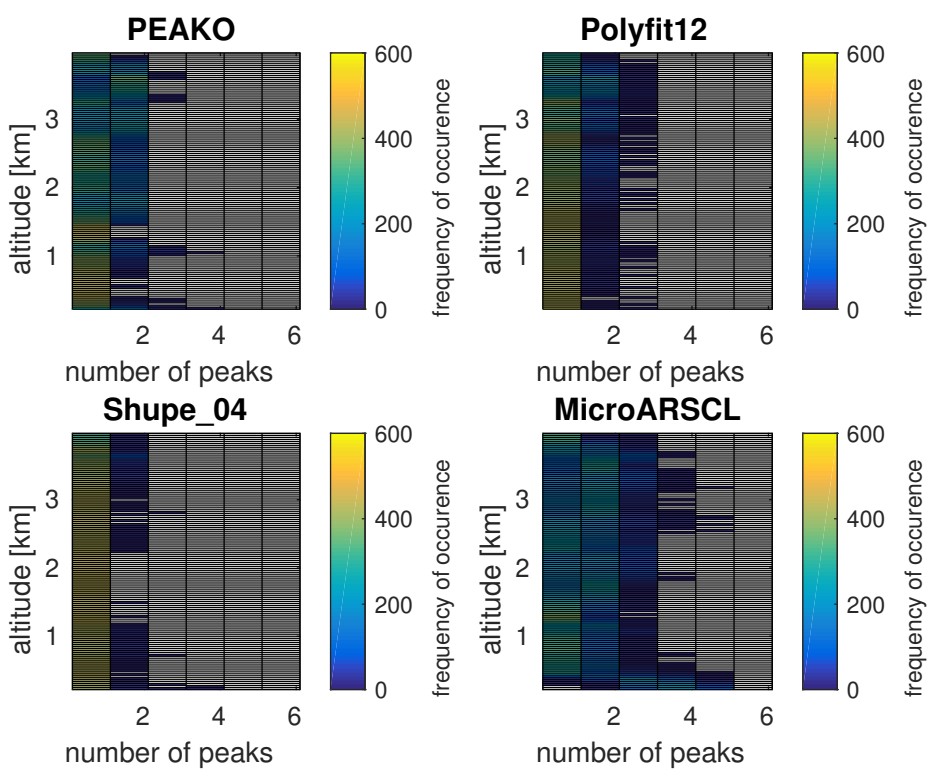

**Figure C3.** Like Fig. 6 but for the case study period on 7 February 2014, 23.75 - 24 UTC in 0.2 - 4 km altitude.




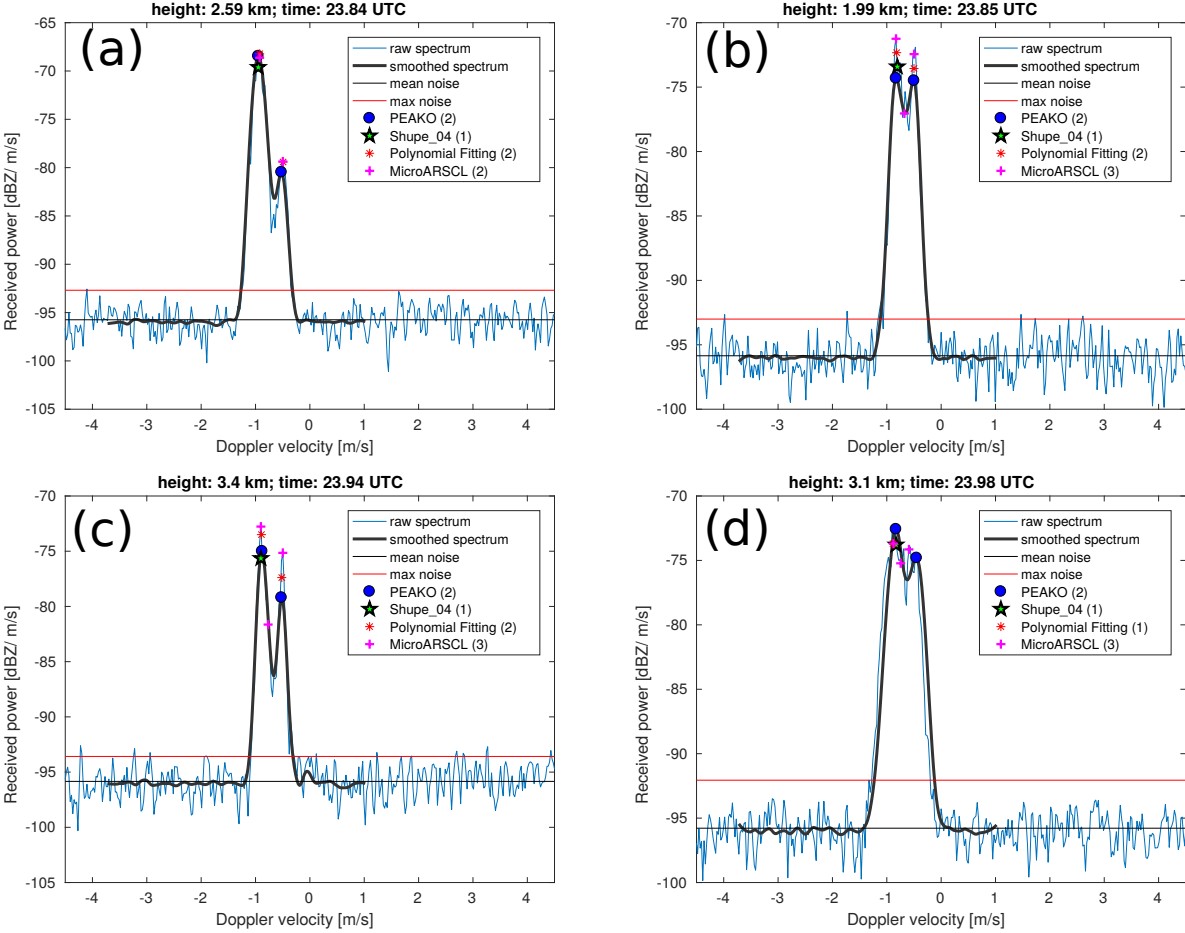

**Figure C4.** Four example spectra picked from the case study on 7 February 2014, 23.75 - 24 UTC. Please note that the y axis scale is different for each of the spectrum plots.



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
