# Peer review of "Development and validation of a supervised machine learning radar Doppler spectra peak finding algorithm"

_Atmospheric Measurement Techniques, 2019_

## Referee Comment (RC1) · Anonymous Referee #1 · 18 May 2019

The paper presents a new method for partitioning cloud-radar Doppler spectra into physically meaningful peaks. There have been many methods adopted over the years for identification of such peaks but these other methods were usually just a step towards an end and not the end itself. The new proposed method is at the heart of this paper. There are interesting ideas in this paper so it is worthy of publication in AMT. Although no show stoppers showed up in reading the paper, there are places where the manuscript can be improved and these are listed in the comments below.

Comments:

0) Mark up on the manuscript made while reading it is contained in the attachment.

[Figure]

Perhaps some of it will be of value to the authors.

1) The essence of the proposed technique is stated in two places within the manuscript: the last full sentence on the bottom of Page 9 which reads "Features prominent enough in time and height to be still visible at all after averaging and smoothing are most probably physical" and the first full sentence at the top of Page 7 which reads "Doppler spectra peaks in low-turbulent liquid cloud droplet layers are very narrow and thus suited to determine the minimum width of a peak considered as physically meaningful." The authors use human analysis of Doppler spectra to identify thresholds in smoothness, peak powers, and peak width to identify physically meaningful peaks. There is some arbitrariness in this approach based on training data. So why not change the approach a bit: use Doppler spectra with no significant returns to identify smoothness, peak power, and peak width thresholds that eliminate all peaks because they are all noise? Any peaks that survive when applied to other observations must then come from hydrometeors. This was the idea that came to mind when reading the two parts of the paper above. It would then come down to a characterization of a radar, much like what the authors hope to do in future studies. Is it a worthwhile approach?

2) The current approach only identifies underlying hydrometeors when they produce separated peaks. The current approach does not work in identifying hydrometeors when the peaks they produce merge together to produce single peaks with or without shoulders. This needs to be pointed out because there are lots of hydrometeors out there that do not produce separate peaks.

3) Perhaps most importantly, the method itself needs to be perfectly described so that it is reproducible. Some might find reproducing the method difficult based on the current description, especially of how the spectra are initially smoothed. Manuscript lines 1-9 on Page 6 are hard to understand in this regard. If the spectra have a temporal resolution of 2 s, then it would take 8 consecutive spectra to cover a 16-s time window. Counting the current spectrum itself, it would take 4 spectra before and 3 after (or 3 before and 4 after) to cover the 16-s time window. Perhaps more likely, 4 spectra before and after the current one were used? If so, these would total 16 s and, together with the current one in the average, a total of 18 s worth of spectra would be averaged. Is this correct? Either way, make perfectly clear what was done. This is a small detail but an important one.

The description of the span in the first two sentences of Page 6 was also hard to understand. First, the phrase "to be considered for spectral smoothing" does not mean that the spectra were actually smoothed. But perhaps the smoothing routine was applied to all spectra in chunks dictated by the span? And the statement "Spectral smoothing is performed using local regression using weighted linear least squares and a 2nd degree polynomial model (loess)." is a bit ambiguous too. How are the local linear and 2nd degree polynomial fits related to each other and to the span? Perhaps a section that illustrates how all of this smoothing works on input spectra to generate an output spectrum would take care of these ambiguities in describing the smoothing. Perhaps it could go something like this: "First, the raw spectrum at the current time and height is replaced by an average spectrum obtained by averaging 27 spectra, 9 in time and 3 in height centered on the current one. Then the averaged spectrum is further smoothed over chunks of spectral bins determined by the span. [Keep going to describe how the span, local linear fits, and 2nd degree polynomial fits work together.]"

Finally, what does "loess" mean? In most dictionaries it is defined as a loamy sediment so not sure what it means in this context.

4) The red lines in all of the Doppler spectra figures represent the "maximum noise". But the definition of the maximum noise is never presented. Is it the maximum value of the power in a raw (unaveraged) spectrum among those Doppler spectral bins identified as noise by the Hildebrand and Sekhon technique? This is what it seems to be based on the pictures. This needs to be clear in the manuscript.

5) Comment 2) above raises another point. What are your criteria (or perhaps criterion) for eliminating spectral peaks that are too low in power? Are peaks below the maximum noise obtained from the raw spectrum allowed? What about peaks below the maximum noise obtained from the averaged spectrum? This needs to be clear.

6) The current method seems to have two separate parts: smoothing and peak identification. The caption to Figure 7 indicates that the peak prominence and peak width thresholds were applied to the incoherently averaged and span-smoothed spectra whereas MicroARSCL and perhaps the other algorithms were not. It would be interesting to assess the importance of smoothing for all of the algorithms involved. To this end applying all four algorithms to the raw spectra and then to the incoherently averaged and span-smoothed spectra is of interest. At least all of the algorithms, and especially MicroARSCL, should be applied to the smoothed spectra and then compared. Doing so would help to differentiate the impacts of smoothing versus feature identification.

7) The word "well" appears on line 13 of the abstract. Replace this subjective statement with something quantitative.

8) Not sure what AMT guidelines are, but using past tense to describe events that happened in the past is perhaps preferable to using present tense in such descriptions. Same goes for describing what one did to pursue the study.

9) Page 4, Lines 22-24: These lines describe levels of training and testing. This information is not carried through to Table 1. Table 1 and Lines 22-24 need to be strongly coupled in terms of wording as this would make clearer what data were used for which purpose.

10) Page 7, Line 8: Figure 3 does not contain any purely red circles, but rather red dots surrounded by blue circles.

15) Starting around Figure 9, the figures are referenced out of order. Not sure what AMT guidelines are, but referencing figures in order makes them easy to find.

16) Figures 8 and 9 should be identically formatted in every way to make their comparison as easy as is possible.

17) A Doppler spectra peak identification procedure built for wind profilers might be applicable to cloud-radar Doppler spectra. It is based on fuzzy logic and received a fair amount of development effort:

Cornman et al. (1998) A Fuzzy Logic Method for Improved Moment Estimation from Doppler Spectra. Journal of Atmospheric and Oceanic Technology, 15, 1287-1305.

This approach has the attribute that thresholds on any one variable do not have to be fixed.

Please also note the supplement to this comment:
https://www.atmos-meas-tech-discuss.net/amt-2019-48/amt-2019-48-RC1-supplement.pdf
* * *
[Figure]

**Supplement:**

[revised manuscript text omitted]

---

## Referee Comment (RC2) · Anonymous Referee #2 · 30 May 2019

This manuscript submission describes a new proposed algorithm to find Doppler velocity spectral peaks using an automated methodology. This algorithm purports to more consistently and accurately identify peaks due to different hydrometeor populations - a sometimes difficult task in mixed phase clouds. The authors, in my opinion, present a convincing case that the new machine learning algorithm outperforms previous Doppler spectral peak identifier algorithms. Example cases are shown in the manuscript, with further examples presented as supplementary material. I offer a few minor comments below that will hopefully improve the manuscript.
* * *
[Figure]

1. Abstract, lines 12-14: The authors state that "The new algorithm is found to perform well." "Well" is a very subjective description. A more quantitative descriptor, or at least less subjective language, is preferred. Possible alternative wording that combines two sentences:

The new algorithm consistently identifies Doppler spectra peaks and outperforms other algorithms by reducing noise and increasing temporal and height consistency in detected features.

2. Lines 23-25: Suggest changing nominalized language for stronger writing:

The first step towards characterizing hydrometeor types is determining the number of different populations within a certain cloud volume.

3. Introduction, Lines 21-31: This section of the introduction is very fragmented. The authors inject various Doppler spectra analysis studies in a somewhat non-coherent manner. Maybe it's just a spatial issue (i.e., the authors indent the various studies as stand-along paragraphs comprised of 1-3 sentences). One way to mitigate this issue to allow the science themes, rather than the referenced studies, to drive the content. I envision these lines recast in terms of scientific topic that would allow a more natural flow to the discussion. A suggestion:

Other studies have utilized Doppler spectra analyses to identify cloud microphysical composition and cloud processes operating in Arctic clouds. For instance, four Arctic cloud hydrometeor populations (background ice, cloud, drizzle and new ice) were successfully classified using continuity of spectral modes in time and height combined with high spectral resolution lidar (HSRL) and in-situ observations (Verlinde et al. 2013). BAECC {what does the BAECC acronym represent?!} field campaign analyses have also distinguished up to three noise-floor separated peaks in the recorded Doppler spectra for frontal snow falling through a supercooled water layer (SWL) that produced rimed snowflakes (Kalesse et al 2016). These respective peaks were then used to track microphysical processes along slanted fall streaks, although this documented case was

special due to the separation of peaks by the noise-floor (merged peaks are usually observed, motivating the need to develop robust cloud radar Doppler spectrum peak separation techniques). Finally, KAZR observations of liquid-only and mixed-phase clouds at Oliktok Point, Alaska have been used to identify multiple Doppler peaks using the depth of the local minimum between the main peak and sub-peak as the main separation criteria (Williams et al. 2018).

All these efforts, using somewhat differing approaches, show that there is a need. . . [continue with the rest of the content from the last introductory paragraph].

I also suggest adding a final sentence to the introduction that briefly introduces what the current study will accomplish. For example, "This study describes a new algorithm that adopts machine learning tools to classify Doppler spectra peaks in complex mixed phase cloud scenarios" – or something similar to this statement that properly whets the readers' appetites.

4. Figure 1: Are these truly random spectra chosen from 16 February 2014? Or are they neighboring spectra, where neighboring can be defined as either spatial (height) or temporal?

5. Page 5, Line 2: 21 February 21 2014 -> 21 February 2014

6. Page 6, Lines 3-4: How did the chosen smoothing method produce the most promising results? Is there any quantitative measure to optimally select the smoothing method (like line fitting parameters)?

7. Page 6, Lines 5-9: Was there a compelling reason to choose the 16 s temporal and 90 m spatial smoothing parameters? This question is probably related to the previous comment. The obvious answer is that spatiotemporal smoothing needs to capture the multi-modal peaks shown in Fig. 2 without completely smearing out the features. I guess I'm having a difficult time being convinced that one could empirically derive the best smoothing parameters and method based only on an "eye test" without further

quantitative support.

Post-hoc comment: The appendix content nicely lends further support for how the algorithm works with the adopted spatiotemporal constraints. I was initially going to suggest appendix material that shows how the algorithm would perform with different smoothing methods and parameters - maybe include a final brief appendix section illustrating the sensitivity of one or two cases to different smoothing schemes or spatiotemporal averaging parameters?

8. Figure 8 caption: I recommend adding what the black dashed line indicates. It is obviously the SLW layer that is again repeated in a later figure, but it should probably be mentioned here, too.
* * *

---

## Author Comment (AC1) · 4 Jul 2019

**Reply to comments of Reviewer # 1:**

We want to thank Reviewer 1 for carefully reading our manuscript and the helpful comments. We are addressing the raised comments in a point-by-point way below:

0) Mark up on the manuscript made while reading it is contained in the attachment. Perhaps some of it will be of value to the authors.

Reply: The notes were certainly of use and we have followed many of the handwritten hints. Thank you for making the extra effort.

1) The essence of the proposed technique is stated in two places within the manuscript: the last full sentence on the bottom of Page 9 which reads "Features prominent enough in time and height to be still visible at all after averaging and smoothing are most probably physical" and the first full sentence at the top of Page 7 which reads "Doppler spectra peaks in low-turbulent liquid cloud droplet layers are very narrow and thus suited to determine the minimum width of a peak considered as physically meaningful." The authors use human analysis of Doppler spectra to identify thresholds in smoothness, peak powers, and peak width to identify physically meaningful peaks. There is some arbitrariness in this approach based on training data. So why not change the approach a bit: use Doppler spectra with no significant returns to identify smoothness, peak power, and peak width thresholds that eliminate all peaks because they are all noise? Any peaks that survive when applied to other observations must then come from hydrometeors. This was the idea that came to mind when reading the two parts of the paper above. It would then come down to a characterization of a radar, much like what the authors hope to do in future studies. Is it a worthwhile approach?

Reply:
The reviewer criticises the arbitrariness of human-created data to train the algorithm, and suggests to change the technique, not using peaks marked by users in Doppler spectra but spectra with no significant peaks at all as input data, to establish thresholds eliminating all noise. These are several good points that deserve a more detailed discussion.

1. We agree that there is arbitrariness in the approach using peaks detected by humans as training to the algorithm. But even though being by definition subjective, the human brain is perfectly tuned to pattern recognition and a role model to many machine learning applications. We tried to counter possible problems caused by human subjectivity by having the training data created by different experienced radar scientists.

2. The suggested alternative approach is definitely an interesting idea, however would not change our technique by 'a bit' but quite drastically. The phrase "Features prominent enough in time and height to be still visible at all after averaging and smoothing are most probably physical" was possibly misworded. The PEAKO algorithm does not primarily aim at determining whether there is physically meaningful signal in a spectrum or not, but at separating several meaningful signals within one spectrum. If this application can be trained using spectra with no significant returns at all is questionable. Usually, the problem is not to detect the main (primary) peak but to distinguish merged peaks. We changed the phrasing of the last sentence on page 9 to "Maxima prominent enough in time and height to be still visible after averaging and smoothing are most probably physical".

3. Eliminating noise is an aspect which is taken care of in part already by applying the Hildebrand & Sekhon criterion to the spectra. No peaks with magnitude below the noise threshold determined by their approach can be detected by PEAKO.

2) The current approach only identifies underlying hydrometeors when they produce separated peaks. The current approach does not work in identifying hydrometeors when the peaks they produce merge together to produce single peaks with or without shoulders. This needs to be pointed out because there are lots of hydrometeors out there that do not produce separate peaks.

Reply: True. The algorithm can only detect a hydrometeor population if it constitutes a maximum in the Doppler spectrum. We added the following sentence to the conclusions, which hopefully stresses this aspect more: "The described approach only identifies underlying hydrometeor populations if the particle types differ sufficiently in their terminal fall velocities to produce individual Doppler spectrum peaks."

3) Perhaps most importantly, the method itself needs to be perfectly described so that it is reproducible. Some might find reproducing the method difficult based on the current description, especially of how the spectra are initially smoothed. Manuscript lines 1-9 on Page 6 are hard to understand in this regard. If the spectra have a temporal resolution of 2 s, then it would take 8 consecutive spectra to cover a 16-s time window. Counting the current spectrum itself, it would take 4 spectra before and 3 after (or 3 before and 4 after) to cover the 16-s time window. Perhaps more likely, 4 spectra before and after the current one were used? If so, these would total 16 s and, together with the current one in the average, a total of 18 s worth of spectra would be averaged. Is this correct? Either way, make perfectly clear what was done. This is a small detail but an important one.
The description of the span in the first two sentences of Page 6 was also hard to understand. First, the phrase "to be considered for spectral smoothing" does not mean that the spectra were actually smoothed. But perhaps the smoothing routine was applied to all spectra in chunks dictated by the span? And the statement "Spectral smoothing is performed using local regression using weighted linear least squares and a 2nd degree polynomial model (loess)." is a bit ambiguous too. How are the local linear and 2nd degree polynomial fits related to each other and to the span? Perhaps a section that illustrates how all of this smoothing works on input spectra to generate an output spectrum would take care of these ambiguities in describing the smoothing. Perhaps it could go something like this: "First, the raw spectrum at the current time and height is replaced by an average spectrum obtained by averaging 27 spectra, 9 in time and 3 in height centered on the current one. Then the averaged spectrum is further smoothed over chunks of spectral bins determined by the span. [Keep going to describe how the span, local linear fits, and 2nd degree polynomial fits work together.]"

Finally, what does "loess" mean? In most dictionaries it is defined as a loamy sediment so not sure what it means in this context.

Reply: You are perfectly right about the time window: We used 4 spectra before and after the current one, totaling 18 s averaging time.
We also agree that the section describing averaging and smoothing of the Doppler spectra is not very well readable. According to your suggestions, we changed the section describing the algorithm. Now it reads:
"As a first step, the raw spectrum at the current time and height is replaced by an average spectrum obtained by averaging 27 spectra, 9 in time and 3 in height centered on the current one. For the given KAZR time-height resolution of 2 s (time) and 30 m (range), this translates to averaging of 18 s in the temporal and 90 m in the spatial dimension. With hydrometeor populations usually appearing in

distinct layers, which are persistent over a certain period of time, more neighbors in time than height are used for averaging. Fig 9a) in Bühl et al. (2016) shows minimum liquid layer depths on the order of 50 to 100 m equivalent to 2-3 range gates assuming 30 m vertical range gate spacing, which motivated our choice of 90 m. The averaged spectrum is then further smoothed using local polynomial regression. The smoothing method applied, locally estimated scatterplot smoothing (loess), performs weighted linear least-square fitting on consecutive sub-sets of adjacent data points with a $2^{nd}$ degree polynomial model. The span for smoothing is the fraction of the total number of data points (here: Doppler bins) of one Doppler velocity spectrum to be used for each local fit. loess smoothing was chosen empirically after testing different methods because it showed the best ability to capture peaks while filtering out noise. The span is varied in a range between 3.5 % and 13 %, regularly spaced with a distance of 0.5 %."

4) The red lines in all of the Doppler spectra figures represent the "maximum noise". But the definition of the maximum noise is never presented. Is it the maximum value of the power in a raw (unaveraged) spectrum among those Doppler spectral bins identified as noise by the Hildebrand and Sekhon technique? This is what it seems to be based on the pictures. This needs to be clear in the manuscript.

Reply: We added "Concerning the highest peak of the Doppler spectrum, the prominence is the power difference between the peak maximum and the mean of the spectral noise determined by Hildebrand and Sekhon (1974)." and "The red horizontal line marks the maximum value of the power in the raw (blue) spectrum among those Doppler spectral bins identified as noise by Hildebrand and Sekhon (1974). The black horizontal line is drawn at the mean power of the Doppler bins containing only noise." to the caption of Figure 2.

5) Comment 2) above raises another point. What are your criteria (or perhaps criterion) for eliminating spectral peaks that are too low in power? Are peaks below the maximum noise obtained from the raw spectrum allowed? What about peaks below the maximum noise obtained from the averaged spectrum? This needs to be clear.

Reply: No peaks below the maximum noise obtained from the raw spectrum can be detected by PEAKO. The noise obtained from the averaged spectrum is not considered at all for peak identification. To make this more clear to the reader, we added the following description:
"In the next step, local maxima are identified in the averaged and smoothed spectrum. Only peaks with powers above the raw spectrum's maximum noise are considered. Finally, peaks with prominences below the prominence threshold and widths smaller than the minimum peak width are excluded."

6) The current method seems to have two separate parts: smoothing and peak identification. The caption to Figure 7 indicates that the peak prominence and peak width thresholds were applied to the incoherently averaged and span-smoothed spectra whereas MicroARSCL and perhaps the other algorithms were not. It would be interesting to assess the importance of smoothing for all of the algorithms involved. To this end applying all four algorithms to the raw spectra and then to the incoherently averaged and span-smoothed spectra is of interest. At least all of the algorithms, and especially MicroARSCL, should be applied to the smoothed spectra and then compared. Doing so would help to differentiate the impacts of smoothing versus feature identification.

Reply: Indeed, but here the focus is on comparing existing algorithms with the new PEAKO algorithm, not on adjusting existing ones (e.g. to the same smoothing as is performed in PEAKO). Differences, especially to MicroARSCL, highlight the importance of smoothing for peak identification. In order to assess the importance of smoothing (as opposed to feature identification), we added an Appendix (D)

giving results of a sensitivity study applying e.g. smoothing or not; varying the temporal and spatial averaging and using different smoothing methods:
"To assess the influence of different smoothing schemes and spatiotemporal averaging space on the algorithm's performance, a sensitivity study was performed. Two smoothing methods available in Matlab are the *moving average* and the locally weighted scatterplot smoothing (*lowess*) schemes. *Lowess* smoothing is very similar to *loess* smoothing with the difference that *lowess* utilizes a first-degree polynomial which is fit to the data subset defined by span.

We trained PEAKO in different configurations using the first training dataset (Table 1). The PEAKO configurations tested were the following:

- Averaging over five spectra in temporal and five spectra in spatial scale, which results in an averaging time scale of 10 s and an averaging height of 150 m. The average spectrum is smoothed using the *loess* method.
- Omitting time-height averaging altogether prior to smoothing the spectra using *loess* smoothing.
- Keeping the spatiotemporal averaging fixed at the default of 16 s and 90 m but using *moving average* smoothing instead of the *loess* method
- Keeping the spatiotemporal averaging scale fixed at the default and using *lowess* smoothing instead of *loess* smoothing

The optimized parameters obtained after training PEAKO in each of above listed configurations were applied to the case study presented in Fig. 5. Figure D1 shows the results.
The panels in Fig. D1 all display a similar pattern with respect to peak number. This is not surprising because the training process of PEAKO is the same for each of the methods, i.e. the three adjustable parameters are adjusted to obtain the best agreement with the human-created training data. A change in the spatiotemporal averaging scale towards more neighbors in height and less neighbors in time does not alter the result significantly. However, performing time-height averaging prior to smoothing at all is important as can be seen in the third panel in Fig. D1: If no spatiotemporal averaging is carried out before smoothing, the features detected by PEAKO become less coherent and more noisy. The two lower panels in Fig. D1 explore the effect of different smoothing schemes on the algorithm performance. Both *moving average* and *lowess* methods are able to reproduce the features detected by PEAKO in the default configuration only with some minor deviations."

[Figure]

*Figure D1. Number of Doppler spectrum peaks detected by PEAKO in five different configurations for the selected case study on 2014-02-21 from 22.54 to 22.77 UTC in 2 to 6 km altitude. Top to bottom: Number of peaks detected by PEAKO in the default configuration (16 s temporal and 90 m spatial averaging prior to loess smoothing), this plot is equivalent to the top panel in Fig. 5; number of peaks detected using 10 temporal and 150 m spatial averaging followed by loess smoothing; number of peaks detected without time-height averaging prior to loess smoothing; number of peaks detected using 16 s and 150 m time-height averaging followed by smoothing using the moving average method; number of peaks detected using 6 s and 150 m time-height averaging followed by lowess smoothing*

7) The word "well" appears on line 13 of the abstract. Replace this subjective statement with something quantitative.

Reply: We agree that "well" is subjective and should be replaced. This issue was also raised by Reviewer #2. According to their suggestion, we rephrased this part of the abstract as follows: "The new algorithm consistently identifies Doppler spectra peaks and outperforms other algorithms by reducing noise and increasing temporal and height consistency in detected features."

8) Not sure what AMT guidelines are, but using past tense to describe events that happened in the past is perhaps preferable to using present tense in such descriptions. Same goes for describing what one did to pursue the study.

Reply: Agreed. We decided to stick to the past tense consistently throughout the manuscript when describing what was done in the data analysis.

9) Page 4, Lines 22-24: These lines describe levels of training and testing. This information is not carried through to Table 1. Table 1 and Lines 22-24 need to be strongly coupled in terms of wording as this would make clearer what data were used for which purpose.

Reply: We changed the table so that now three data sets are distinguished, a $1^{st}$ training data set, a $2^{nd}$ training data set and a testing data set, in accordance with the bullet points in Section 3.1 (algorithm description).

10) Page 7, Line 8: Figure 3 does not contain any purely red circles, but rather red dots surrounded by blue circles.

Reply: We rephrased the description of the figure in the text accordingly.

15) Starting around Figure 9, the figures are referenced out of order. Not sure what AMT guidelines are, but referencing figures in order makes them easy to find.

Reply: We moved the first mentioning of Figure 12 to a later paragraph so that figures are referenced in the correct order.

16) Figures 8 and 9 should be identically formatted in every way to make their comparison as easy as is possible.

Reply: We adjusted the grid and y-axis ticks of Figure 9 to match the format of Figure 8

17) A Doppler spectra peak identification procedure built for wind profilers might be applicable to cloud-radar Doppler spectra. It is based on fuzzy logic and received a fair amount of development effort:
Cornman et al. (1998) A Fuzzy Logic Method for Improved Moment Estimation from Doppler Spectra. Journal of Atmospheric and Oceanic Technology, 15, 1287-1305.
This approach has the attribute that thresholds on any one variable do not have to be fixed.

Reply: We thank the reviewer for pointing us to this interesting study. It is certainly true that different approaches for radar peak identification – including fuzzy logic – can be used to study cloud radar Doppler spectra. We acknowledge and cite this reference in the Introduction.

---

## Author Comment (AC2) · 4 Jul 2019

**Reply to comments of Reviewer # 2:**

We want to thank Reviewer 2 for their comments, which certainly aided to improve our manuscript. We are addressing the raised comments in a point-by-point way below:

1. Abstract, lines 12-14: The authors state that "The new algorithm is found to perform well." "Well" is a very subjective description. A more quantitative descriptor, or at least less subjective language, is preferred. Possible alternative wording that combines two sentences:
The new algorithm consistently identifies Doppler spectra peaks and outperforms other algorithms by reducing noise and increasing temporal and height consistency in detected features.

Reply: We agree that a more quantitative wording is preferable to using "well". Thus, we decided to accept the proposed alternative phrasing and combine the two sentences.

2. Lines 23-25: Suggest changing nominalized language for stronger writing:
The first step towards characterizing hydrometeor types is determining the number of different populations within a certain cloud volume.

Reply: According to the suggestion, we adjusted the sentence using the gerund.

3. Introduction, Lines 21-31: This section of the introduction is very fragmented. The authors inject various Doppler spectra analysis studies in a somewhat non-coherent manner. Maybe it's just a spatial issue (i.e., the authors indent the various studies as stand-along paragraphs comprised of 1-3 sentences). One way to mitigate this issue to allow the science themes, rather than the referenced studies, to drive the content. I envision these lines recast in terms of scientific topic that would allow a more natural flow to the discussion. A suggestion:
Other studies have utilized Doppler spectra analyses to identify cloud microphysical composition and cloud processes operating in Arctic clouds. For instance, four Arctic cloud hydrometeor populations (background ice, cloud, drizzle and new ice) were successfully classified using continuity of spectral modes in time and height combined with high spectral resolution lidar (HSRL) and in-situ observations (Verlinde et al. 2013). BAECC {what does the BAECC acronym represent?!} field campaign analyses have also distinguished up to three noise-floor separated peaks in the recorded Doppler spectra for frontal snow falling through a supercooled water layer (SWL) that produced rimed snowflakes (Kalesse et al 2016). These respective peaks were then used to track microphysical processes along slanted fall streaks, although this documented case was special due to the separation of peaks by the noise-floor (merged peaks are usually observed, motivating the need to develop robust cloud radar Doppler spectrum peak separation techniques). Finally, KAZR observations of liquid-only and mixed-phase clouds at Oliktok Point, Alaska have been used to identify multiple Doppler peaks using the depth of the local minimum between the main peak and sub-peak as the main separation criteria (Williams et al. 2018).
All these efforts, using somewhat differing approaches, show that there is a need. . .
[continue with the rest of the content from the last introductory paragraph].

I also suggest adding a final sentence to the introduction that briefly introduces what the current study will accomplish. For example, "This study describes a new algorithm that adopts machine learning tools to classify Doppler spectra peaks in complex mixed phase cloud scenarios" – or something similar to this statement that properly whets the readers' appetites.

Reply: We agree that this section of the text is rather incoherent and gratefully accept the proposed alternative text. This greatly improves the readability of the manuscript. We added the abbreviation for the Biogenic Aerosols – Effects on Clouds and Climate (BAECC) field campaign, which is mentioned in the Abstract for the first time.

4. Figure 1: Are these truly random spectra chosen from 16 February 2014? Or are they neighboring spectra, where neighboring can be defined as either spatial (height) or temporal?

Reply: The caption of Fig. 1 was misworded. The algorithm picks spectra for the user to marks peaks randomly in a previously defined time-height chunk of Doppler spectra. Fig. 1 (center panel) shows one of these randomly picked spectra. The spectra displayed in Fig. 1 in the panels around the center plot are neighboring spectra (in time and height). We changed the label of Fig. 1 for clarity to "The surrounding spectra display the spatially and temporally neighboring spectra." and changed from "[…] Data: random spectra of KAZR observed at TMP [...]" to "[…] Data: KAZR spectra observed at TMP [..]".

5. Page 5, Line 2: 21 February 21 2014 -> 21 February 2014

Reply: We adjusted the text accordingly.

6. Page 6, Lines 3-4: How did the chosen smoothing method produce the most promising results? Is there any quantitative measure to optimally select the smoothing method (like line fitting parameters)?

Reply: We picked the smoothing method by visual inspection, testing different smoothing methods which are by default available in the Matlab Signal Processing Toolbox. The loess-smoothing yielded the best results at separating peaks in a quick sensitivity study. We re-phrased the text to "This smoothing method was chosen empirically after testing different methods since it showed the best ability to capture peaks while filtering out noise". Please also see the reply to the next comment for a more complete answer to the question.

7. Page 6, Lines 5-9: Was there a compelling reason to choose the 16 s temporal and 90 m spatial smoothing parameters? This question is probably related to the previous comment. The obvious answer is that spatiotemporal smoothing needs to capture the multi-modal peaks shown in Fig. 2 without completely smearing out the features. I guess I'm having a difficult time being convinced that one could empirically derive the best smoothing parameters and method based only on an "eye test" without further quantitative support.
Post-hoc comment: The appendix content nicely lends further support for how the algorithm works with the adopted spatiotemporal constraints. I was initially going to suggest appendix material that shows how the algorithm would perform with different smoothing methods and parameters - maybe include a final brief appendix section illustrating the sensitivity of one or two cases to different smoothing schemes or spatiotemporal averaging parameters?

Reply: Averaging is performed over more neighboring spectra in time than in space, because Doppler spectra features, e.g. liquid peaks, occur usually in layers and are more consistent in time than in height. E.g. Figure 9a in Bühl et al. (2016, https://doi.org/10.5194/acp-16-10609-2016) shows minimum liquid layer depth was on the order of 50-100 m equivalent to 2-3 range gates assuming 30 m vertical range gate spacing, which motivates our choice of 90 m. We added the following sentence to the methods description to make this clear to the reader: "Fig 9a) in Bühl et al. (2016) shows minimum

liquid layer depths on the order of 50 to 100 m equivalent to 2-3 range gates assuming 30 m vertical range gate spacing, which motivated our choice of 90 m."

Following the recommendation, we added another appendix section (Appendix D) which assesses the influence of different smoothing schemes and spatiotemporal averaging on the performance of the algorithm:

"To assess the influence of different smoothing schemes and spatiotemporal averaging space on the algorithm's performance, a sensitivity study was performed. Two smoothing methods available in Matlab are the *moving average* and the locally weighted scatterplot smoothing (*lowess*) schemes. *Lowess* smoothing is very similar to *loess* smoothing with the difference that *lowess* utilizes a first-degree polynomial which is fit to the data subset defined by span.

We trained PEAKO in different configurations using the first training dataset (Table 1). The PEAKO configurations tested were the following:

- Averaging over five spectra in temporal and five spectra in spatial scale, which results in an averaging time scale of 10 s and an averaging height of 150 m. The average spectrum is smoothed using the *loess* method.
- Omitting time-height averaging altogether prior to smoothing the spectra using *loess* smoothing.
- Keeping the spatiotemporal averaging fixed at the default of 16 s and 90 m but using *moving average* smoothing instead of the *loess* method
- Keeping the spatiotemporal averaging scale fixed at the default and using *lowess* smoothing instead of *loess* smoothing

The optimized parameters obtained after training PEAKO in each of above listed configurations were applied to the case study presented in Fig. 5. Figure D1 shows the results.

The panels in Fig. D1 all display a similar pattern with respect to peak number. This is not surprising because the training process of PEAKO is the same for each of the methods, i.e. the three adjustable parameters are adjusted to obtain the best agreement with the human-created training data. A change in the spatiotemporal averaging scale towards more neighbors in height and less neighbors in time does not alter the result significantly. However, performing time-height averaging prior to smoothing at all is important as can be seen in the third panel in Fig. D1: If no spatiotemporal averaging is carried out before smoothing, the features detected by PEAKO become less coherent and more noisy. The two lower panels in Fig. D1 explore the effect of different smoothing schemes on the algorithm performance. Both *moving average* and *lowess* methods are able to reproduce the features detected by PEAKO in the default configuration only with some minor deviations."

[Figure]

*Figure D1. Number of Doppler spectrum peaks detected by PEAKO in five different configurations for the selected case study on 2014-02-21 from 22.54 to 22.77 UTC in 2 to 6 km altitude. Top to bottom: Number of peaks detected by PEAKO in the default configuration (16 s temporal and 90 m spatial averaging prior to loess smoothing), this plot is equivalent to the top panel in Fig. 5; number of peaks detected using 10 temporal and 150 m spatial averaging followed by loess smoothing; number of peaks detected without time-height averaging prior to loess smoothing; number of peaks detected using 16 s and 150 m time-height averaging followed by smoothing using the moving average method; number of peaks detected using 6 s and 150 m time-height averaging followed by lowess smoothing*

8. Figure 8 caption: I recommend adding what the black dashed line indicates. It is obviously the SLW layer that is again repeated in a later figure, but it should probably be mentioned here, too.

Reply: We added the following sentence to the figure caption: "The black dashed line marks the boundary of the supercooled liquid layer, indicated by high backscatter and low depolarisation ratio."